

# Exploring the impacts of anthropogenic emission sectors on PM$_{2.5}$ and human health in South and East Asia

Carly L. Reddington[1], Luke Conibear[1], Christoph Knote[2], Ben J. Silver[1], Steve R. Arnold[1], Dominick V. Spracklen[1]

[1]Institute for Climate & Atmospheric Science, School of Earth and Environment, University of Leeds, Leeds, UK
[2]Meteorological Institute, LMU Munich, Munich, Germany

*Correspondence to*: C. L. Reddington (c.l.s.reddington@leeds.ac.uk)

**Abstract.** To improve poor air quality in Asia and inform effective emission-reduction strategies, it is vital to understand the contributions of different pollution sources and their associated human health burdens. In this study, we use the WRF-Chem

regional atmospheric model to explore the air quality and human health benefits of eliminating emissions from seven different anthropogenic sectors (transport, industry, shipping, agriculture, energy generation, residential combustion and open biomass burning) over South and East Asia in 2014. We evaluate WRF-Chem against measurements from air quality monitoring stations across the region and find the model captures the spatial distribution and magnitude of PM$_{2.5}$ (particulate matter < 2.5 μm diameter). We find that eliminating emissions from residential energy use, industry or open biomass burning yield the

largest reductions in population-weighted PM$_{2.5}$ concentrations across the region. The largest human health benefit is achieved by eliminating either residential or industrial emissions, averting 467,000 (409,000-542,000) or 283,000 (95UI: 226,000-358,000) annual premature mortalities, respectively in India, China and Southeast Asia; with fire prevention averting 28,000 (95UI: 24,000-32,000) annual premature mortalities across the region. We compare our results to previous sector-specific emission studies. Across these studies, residential emissions are the dominant cause of particulate pollution in India, with a

multi-model mean contribution of 42% to population-weighted annual mean PM$_{2.5}$. Residential and industrial emissions cause the dominant contributions in China, with multi-model mean contributions of 29% for both sectors to population-weighted annual mean PM$_{2.5}$. Future work should focus on identifying the most effective options within the residential, industrial and open biomass burning emission sectors to improve air quality across South and East Asia.

## 1 Introduction

Rapid industrialisation and urbanisation combined with slow implementation of environmental legislation and clean residential fuels has led to serious air quality problems across Asia. Exposure to poor air quality is associated with detrimental acute and chronic health effects, including premature mortality due to cardiopulmonary diseases and lung cancer (Burnett et al. 2014; Cohen et al. 2017), and reduced life expectancy (Apte et al., 2018). Specifically, exposure to ambient fine particulate matter (with diameters < 2.5 μm; PM$_{2.5}$) pollution is a leading risk factor for human health in Asia and is estimated to cause around



1 million premature deaths every year in both China and India (The Global Burden of Diseases, Injuries, and Risk Factors Study 2016 (GBD2016); Cohen et al., 2017; Li et al., 2018; Burnett et al., 2018).

In China, the government have begun to tackle these air quality problems in recent years by introducing policies to reduce air pollutant emissions. Satellite and ground-based measurements indicate that concentrations of some air pollutants ($PM_{2.5}$ and
sulphur dioxide ($SO_2$)) have begun to decline in China within the last decade (Ma et al., 2016; van der A et al., 2017; Lin et al., 2018; Silver et al., 2018). India is also introducing policies aimed at addressing the health burden from air pollution (Sagar et al 2016; Goldemberg et al 2018). Many of these policies are due to be unified within the upcoming National Clean Air Programme (NCAP) to provide a framework for air quality management with the aim of attaining Indian air quality standards (Ministry of Environment Forests and Climate Change, 2018). However, despite these policies being introduced in China and
India, ambient $PM_{2.5}$ pollution remains a problem in both countries, with measured annual mean concentrations well in excess of the World Health Organization (WHO) Air Quality Guideline concentration of 10 µg m$^{-3}$ (Brauer et al., 2016; Yang et al., 2018; Silver et al., 2018).

To improve poor air quality in Asia and inform effective emission-reduction strategies, it is vital to understand the major contributing sources and processes that to lead to poor air quality and associated human health effects. Policies that have been
implemented in North America and Europe to improve air quality may have limited effectiveness in Asia due to differences in emission sources. Therefore, there is a strong need for new research on source contributions specifically focussed on countries in Asia.

To quantify source contributions to $PM_{2.5}$ and other air pollutants at a regional or national level, atmospheric chemistry-transport models can be applied (e.g. Ying et al., 2014; Hu et al., 2015; Wang et al., 2015; Shi et al., 2017; Timmermans et al.,
2017; Qiao et al., 2018) using two main methods. The first method uses a "tagging" approach (also referred to as a "source-attribution" or "source-oriented" approach), where species in the model are tagged to trace the origin of the air pollutant of interest. This technique allows accurate quantification of the contributions of specified emission sources, model process and/or source regions to a given air pollutant. The second method uses a "removal" approach (also referred to as a "source-subtraction" approach or "sensitivity analysis") where multiple model simulations are performed with different emission
source-sectors or source-regions excluded ("zeroed out" or "switched off"). The effective contribution of the source of interest is calculated as the difference in simulated pollutant concentrations between the perturbed simulation and a control simulation (including all sources).

If the behaviour of air pollutants from emission to atmospheric concentration was linear, these two methods would yield the same results. However, the processing and resulting concentrations of certain air pollutants, particularly secondary pollutants
(i.e. those partially or exclusively formed in the atmosphere), can be highly non-linear. Following this, the "removal" modelling approach allows accurate quantification of the change in past, current or future air pollutant concentrations should the specified emission sector be eliminated or reduced as a result of emission control strategies or other reasons. This approach is better





suited to test the results of implementing planned or suggested emission controls on air pollutant concentrations than the "tagging" approach.

Using the "tagging" approach, Shi et al. (2017), Timmermans et al. (2017) and Qiao et al., (2018) analysed the source apportionment of $PM_{2.5}$ across China. These studies consistently identified residential combustion and industry as the main

contributing emission sectors to $PM_{2.5}$ with some disagreement regarding the importance of the transport sector. Karagulian et al. (2017) used the "removal" approach and also found the largest relative contributions to $PM_{2.5}$ in China were from the industrial and residential sectors, with the residential sector dominating contributions in India.

By combining atmospheric chemistry-transport models with exposure-response functions (from e.g. Burnett et al. (2014)), several studies have quantified the disease burden associated with exposure to ambient $PM_{2.5}$ from different emission sectors

either globally (e.g. Lelieveld et al., 2015; Butt et al., 2016; Silva et al., 2016; Liang et al., 2018) or specifically for India and/or China (Archer-Nicholls et al., 2016; Global Burden of Disease from Major Air Pollution Sources (GBD-MAPS), 2016; 2018; Hu et al., 2017; Aunan et al., 2018; Gao et al., 2018; Gu et al., 2018; Upadhyay et al., 2018; Guo et al., 2018; Conibear et al., 2018a) and Southeast Asia (Koplitz et al., 2017). Studies that consider contributions from multiple emission sectors, generally find that $PM_{2.5}$-related health effects are dominated in India by emissions from residential energy use (Lelieveld et al., 2015;

Silva et al., 2016; GBD-MAPS, 2018; Upadhyay et al., 2018; Guo et al., 2018; Conibear et al., 2018a) and in China by emissions from residential energy use (Lelieveld et al., 2015; Silva et al., 2016) or industry (GBD-MAPS, 2016; Hu et al., 2017; Gu et al., 2018). However, the estimates of sectoral contributions to premature mortality from ambient $PM_{2.5}$ exposure vary widely between the studies, largely caused by differences in the applied mortality estimation approaches ("attribution" or "substitution"; Conibear et al., 2018a), exposure-health impact functions, model processes and structure (including model grid

resolution), anthropogenic emissions data, and population data. It is often challenging to distinguish the different methods used in these studies and to understand the implications of the different methods on the results presented.

The implications of using different approaches for estimating the health burden associated with $PM_{2.5}$ exposure in India was explored and demonstrated recently by Conibear et al. (2018a). Conibear et al. (2018a) found that 52% of population-weighted annual mean $PM_{2.5}$ concentrations and 511,000 (95UI: 340,000-697,000) annual premature mortalities in India were attributed

to residential energy use (the "attribution" approach). However, removing residential emissions would avert only 256,000 (95UI: 162,000-340,000) annual premature mortalities (26% of the total) (the "substitution" approach), due to the non-linear exposure–response relationship causing health effects to saturate at high $PM_{2.5}$ concentrations.

To our knowledge, the potential averted disease burden from eliminating multiple different pollution sources has not yet been quantified specifically for China and Southeast Asia at high resolution. Here we use the source-"removal" and mortality-

"substitution" approaches in a high-resolution regional model (following Conibear et al. (2018a)) to quantify the sector-specific air quality benefit and avoided disease burden in China, Mainland Southeast Asia and the Indian Subcontinent. We focus on



anthropogenic emission sectors (land transport, industry, agriculture, power generation, residential combustion and shipping) and open biomass burning (including agricultural and deforestation fires).

In this paper, we also produce the most comprehensive summary to date of previous studies on sector-specific PM$_{2.5}$ and disease-burden contributions in India and China. We document both the methods used and the results from these previous
studies to enable more informed comparisons between them, and also to develop a multi-model range in estimates of the sectoral contributions to PM$_{2.5}$ and disease burden in India and China.

## 2. Methods

### 2.1 Model description

To simulate regional PM$_{2.5}$ concentrations we used the Weather Research and Forecasting model coupled with Chemistry
(WRF-Chem; Grell et al., 2005) version 3.7.1, which simulates the emission, transport, mixing, chemical transformation and removal of trace gases and aerosol simultaneously with meteorology. We use the same model version and set-up as Conibear et al. (2018a), who give a detailed model description in the methods.

Aerosol physics and chemistry are treated using the Model for Simulating Aerosol Interactions and Chemistry (MOSAIC; Zaveri et al., 2008) scheme, including grid-scale aqueous chemistry and extended treatment of organic aerosol (Hodzic and
15 Jimenez, 2011; Hodzic and Knote, 2014). Four discrete size bins are used within MOSAIC (0.039–0.156 μm, 0.156–0.625 μm, 0.625–2.5 μm, 2.5–10 μm) to represent the aerosol size distribution. Gas-phase chemical reactions are calculated using the chemical mechanism Model for Ozone and Related Chemical Tracers, version 4 (MOZART-4) (Emmons et al., 2010), with several updates to photochemistry of aromatics, biogenic hydrocarbons and other species relevant to regional air quality (Knote et al., 2014).

Simulated mesoscale meteorology is kept in line with analysed meteorology through grid nudging to the National Centre for Environmental Prediction (NCEP) Global Forecast System (GFS) analyses to limit errors in mesoscale transport (NCEP, 2000; 2007). The model meteorology was reinitialised every month to avoid drifting of WRF-Chem and spun up for 12 hours, while chemistry and aerosol fields were retained to allow for pollution build-up and mesoscale pollutant transport phenomena to be captured. During the simulations, horizontal and vertical wind, potential temperature and water vapour mixing ratio were
nudged to GFS analyses in all model layers above the planetary boundary layer. Meteorological conditions were initialised by NCEP GFS 6-hourly analyses at 0.5° resolution. These, together with GFS 3-h forecasts in between were also used for boundary conditions and grid analysis nudging (NCEP, 2000; 2007). MOZART-4/Goddard Earth Observing System Model version 5 (GEOS5) 6-hourly simulation data (NCAR, 2016) were used for chemical and aerosol boundary conditions.

We used two model domains; one over the Indian subcontinent and one over East Asia (including Eastern and Southern China
and Mainland Southeast Asia). Both model domains use a Lambert conformal conical projection with a horizontal resolution of 30 km x 30 km. The model domain over the Indian subcontinent covers a 140x140 grid (Conibear et al., 2018a); while the





model domain over East Asia covers a 130x124 grid. The domains have 33 vertical levels up to a minimum pressure of 10 hPa. We re-gridded the model output, using linear interpolation, onto a regular latitude-longitude grid at 0.25° × 0.25° resolution. The results presented in Sect 3. (including the model evaluation statistics, sectoral contributions to PM$_{2.5}$ and health effects) were all calculated/obtained for the two model domains separately. The two model domains are combined in Fig 1a

for display purposes only (where the domains overlap, the grid cells with the maximum annual mean PM$_{2.5}$ concentrations in the control simulation are shown).

We calculated the contribution of specific emission sectors to PM$_{2.5}$ concentrations using the "removal" approach i.e. by switching off emission sectors one-at-a-time in individual simulations. The emission sectors investigated were agriculture (AGR), power generation (ENE), industrial non-power (IND), residential energy use (RES), land transport (TRA), open

biomass burning (BBU), and shipping (SHP; only in the East Asia domain). All simulations were run for the same time period, with identical reinitialisation intervals for the model meteorology (monthly). The simulation period was for one year from 00:00 9 January 2014 to 23:00 8 January 2015, with the first eight days of January 2014 run as spin-up.

### 2.1.1 Description of emissions inventories

Anthropogenic emissions were taken from the Emission Database for Global Atmospheric Research with Task Force on

Hemispheric Transport of Air Pollution (EDGAR-HTAP) version 2.2 at 0.1°×0.1° horizontal resolution (Janssens-Maenhout et al., 2015). For emissions over Asia EDGAR-HTAPv2.2 uses the Model Intercomparison Study for Asia Phase III (MIX) mosaic Asian anthropogenic emission inventory version 1.0 at 0.25°×0.25° horizontal resolution (Li et al., 2017). For China, MIX uses the Multiresolution Emission Inventory for China (MEIC) developed by Tsinghua University (http://www.meicmodel.org) and a high-resolution ammonia (NH$_3$) emission inventory by Peking University (Huang et al.,

2012) to replace MEIC emissions for NH$_3$ over China. For India, MIX uses the Indian emission inventory provided by Argonne National Laboratory (Lu et al 2011; Lu and Streets, 2012) for sulphur dioxide (SO$_2$), black carbon (BC) and organic carbon (OC) for all sectors as well as nitrogen oxides (NOx) for power plants, and REAS2.1 (Kurokawa et al., 2013) for other species. Gaps in EDGAR-HTAPv2.2 were filled by the bottom-up global emission inventory EDGARv4.3.

The EDGAR-HTAPv2.2 inventory includes emissions of SO$_2$, NOx, carbon monoxide (CO), non-methane volatile organic

compounds (NMVOC), NH$_3$, BC and OC from the following source sectors: aviation, shipping, agriculture, power generation, industrial non-power, land transport and residential energy use. The following descriptions of these emissions sectors are from Janssens-Maenhout et al. (2015). The aviation sector includes all international and domestic aviation. The shipping sector includes all international (marine) shipping but not inland waterways. The industrial sector includes emissions from manufacturing, mining, metal, cement, chemical, and solvent industries. Land transport includes all transport by road, railway,

inland waterways, pipeline and other ground transport of mobile machinery. The agricultural sector includes emissions from livestock and crop cultivation but not from agricultural waste burning or Savannah burning. Emissions from residential energy



include small-scale combustion devices for heating, cooking, lighting and cooling in addition to supplementary engines for residential, commercial, agricultural, solid waste and wastewater treatment.

Daily mean biomass burning emissions were taken from the Fire Inventory from NCAR (FINN) version 1.5, with a spatial resolution of 1 km x 1 km (Wiedinmyer et al., 2011) for the year 2014. Biogenic emissions were calculated online by the

5 Model of Emissions of Gases and Aerosol from Nature (MEGAN; Guenther et al., 2006). Dust emissions were calculated online through the Georgia Institute of Technology-Goddard Global Ozone Chemistry Aerosol Radiation and Transport (GOCART) model with Air Force Weather Agency (AFWA) modifications (LeGrand et al., 2019). Anthropogenic dust emissions (e.g. re-suspended road dust, construction dust etc.) are not included. It is important to note dust emissions may be underestimated across Asia in these simulations (Conibear et al., 2018a).

**2.2 Health impact estimation**

We calculated the disease burden due to exposure to ambient $PM_{2.5}$ using the Integrated Exposure-Response (IER) functions from The Global Burden of Diseases, Injuries, and Risk Factors Study 2015 (GBD2015) with age-specific modifiers for each disease to estimate the relative risk of premature mortality due to exposure to various $PM_{2.5}$ concentrations (GBD2015; Cohen et al., 2017). We estimated the disease burden from lower respiratory infection (LRI) for early, late and post neonatal, and

15 populations between 1 and 80 years upwards in 5-year groupings; and from ischaemic heart disease (IHD), cerebrovascular disease or stroke (STR), chronic obstructive pulmonary disease (COPD) and lung cancer (LC) for adults over 25 years old, split into 5-year age groups. We used the parameter distributions of α, β and γ from the GBD2015 for 1000 simulations to derive the mean IER with 95% uncertainty intervals (GBD2015; Cohen et al., 2017). The IER functions have uniform theoretical minimum risk exposure levels (TMREL) for $PM_{2.5}$ between 2.4–5.9 µg m$^{-3}$. The calculation of the disease burden

and uncertainty is described in further detail in the Supplementary Material (Sect. S1).

As in Conibear et al. (2018a), sector-specific mortality was calculated using the "subtraction" method. The "subtraction" method calculates the sector-specific premature mortality ($M_{SECTOR}$) as the difference between the premature mortality from all sources ($M_{ALL}$) and the premature mortality when one sector has been removed ($M_{SECTOR\_OFF}$) as in Eq. 1:

$$M_{SECTOR} = M_{ALL} - M_{SECTOR\_OFF} \quad (1)$$

We also calculated the sector-specific mortality using the "attribution" method (following Conibear et al. (2018a)) to compare our results with previous studies that used this method. The "attribution" method first calculates the fractional sectoral reduction in $PM_{2.5}$ concentrations from removing an emission sector ($PM_{2.5\_SECTOR\_OFF}$) and then uses this fraction to scale the total premature mortality estimate (Eq. 2).

$$M_{SECTOR} = M_{ALL} (PM_{2.5\_ALL} - PM_{2.5\_SECTOR\_OFF})/PM_{2.5\_ALL} \quad (2)$$

There is large uncertainty associated with calculating the health effects due to exposure to ambient $PM_{2.5}$, with recent studies suggesting that the IER functions may underestimate relative risk (Yin et al., 2017; Li et al., 2018) and/or disease burden



(Burnett et al., 2018). For example, recent epidemiological cohort studies in China suggest that the IER functions may underestimate the relative risk of cause-specific mortality due to long-term exposure to $PM_{2.5}$ for $PM_{2.5}$ concentrations experienced in China and other low- and middle-income countries (Yin et al., 2017; Li et al., 2018). These studies suggest that our premature mortality estimates, at least in China, may be conservative.

The population count (P) data set at $0.25° \times 0.25°$ resolution was obtained from the Gridded Population of the World, Version 4 (GPWv4), created by the Centre for International Earth Science Information Network (CIESIN) and accessed from the National Aeronautics and Space Administration (NASA) Socioeconomic Data and Applications Centre (SEDAC) (GPWv4, 2016). The United Nations adjusted version was implemented for 2015 with total populations of 1.302 billion for India and 1.380 billion for China (1.402 billion for China and Taiwan). The WRF-Chem model domain used in this study (described in
Sect. 2.1) includes 92% of the population of China. Population age composition was taken from the GBD2015 population estimates for 2015 (GBD Collaborative Network, 2016).

### 2.3 $PM_{2.5}$ measurements

To evaluate our model-simulated surface PM concentrations, we used measured annual mean $PM_{2.5}$ and $PM_{10}$ concentrations from the World Health Organization database (2016, 2018). The database consists of city-average $PM_{2.5}$ and $PM_{10}$
concentrations obtained from multiple ground station measurements. Roughly 75% of measurements are from urban areas of at least 20,000 inhabitants, with the remaining 25% from smaller areas of up to 20,000 residents. The years of available measurements range from 2008 to 2016. Some cities in the database only have measurements of $PM_{10}$ concentrations. For these locations, $PM_{2.5}$ concentrations have been calculated by the WHO from the measured $PM_{10}$ concentration using national conversion factors ($PM_{2.5}/PM_{10}$ ratio) either provided by the country or estimated as population-weighted averages of urban-
specific conversion factors (estimated as the mean $PM_{2.5}/PM_{10}$ ratio of stations for the same year) for the country (WHO 2016, 2018). These calculated $PM_{2.5}$ concentrations make up 41% of the measurements used in this study (see Table 1). For $PM_{2.5}$ measurements in Vietnam, we found large differences between measured and converted concentrations and therefore only include measured concentrations in the model evaluation (Sect. 3.1) for this country.

### 2.3.1 Comparing simulated and measured $PM_{2.5}$ concentrations

To evaluate model-simulated annual mean $PM_{2.5}$ concentrations against measurements from the WHO (Sect. 3.3), we selected measurement years to match or to be or close as possible to the simulation year of 2014. The simulated annual mean surface $PM_{2.5}$ concentrations from the control simulation were linearly interpolated to the location of the measurement station, using the longitude and latitude of the central part of the relevant town/city/municipality if the measurement represented an average of multiple stations. To quantify the agreement between model and observations, we use the Pearson correlation coefficient
(r) and normalised mean bias factor (NMBF) as defined by Yu et al. (2006). A positive NMBF indicates the model



overestimates the observations by a factor of NMBF+1. A negative NMBF indicates the model underestimates the observations by a factor of 1–NMBF.

## 3. Results

### 3.1 Model evaluation

The model captures the observed spatial distribution of annual mean $PM_{2.5}$ concentrations, for the year 2014, particularly over China, India, Bangladesh and Thailand (Fig. 1; r=0.55). Figure 1 compares simulated and measured annual mean $PM_{2.5}$ concentrations over the Indian Subcontinent, Mainland Southeast Asia and eastern and southern China. Figure 1a shows that the model simulates high annual mean $PM_{2.5}$ concentrations (~80-160 µg m$^{-3}$) over the Indo-Gangetic Plain in northern India and over the North China Plain and Sichuan Basin regions in China; with lower concentrations simulated over southern and

western India, southern China and Mainland Southeast Asia. The spatial agreement between model and measurements is improved when comparing against 2014 measurements only (r=0.76) or when we compare against measured $PM_{2.5}$ only and discard values converted from $PM_{10}$ (r=0.63).

Over the whole domain, simulated annual mean $PM_{2.5}$ concentrations are unbiased against the WHO measurements (Fig. 1b; NMBF=0.09; equivalent to a factor 1.09 greater than measured values). On average, the model simulates annual mean $PM_{2.5}$

concentrations within a factor 1.5 of the measurements in China (NMBF=0.33; Table 1), Thailand (NMBF=0.06), India (NMBF=-0.05), Bangladesh (NMBF=-0.26), Vietnam (NMBF=0.46) and the Republic of Korea (NMBF=-0.32); and within a factor of 2.3 in Myanmar (NMBF=-1.27), Nepal (NMBF=-0.81) and Bhutan (NMBF=-0.63). The negative model biases (up to a factor of 2.27 underestimation) may be due to underestimation of open biomass burning and anthropogenic emissions in some regions. Simulated $PM_{2.5}$ concentrations and thus the estimated $PM_{2.5}$-related disease burdens for countries with negative

model biases are likely to be conservative.

For annual mean $PM_{2.5}$ concentrations above ~60 µg m$^{-3}$ in China, the model is positively biased against the measurements; this may be due to using anthropogenic emissions data from 2010 and comparing with measurements from 2014. $PM_{2.5}$ emissions, particularly those in the industrial and power generation sectors, are reported to have decreased across China between 2010 and 2014 (Zheng et al., 2018). It should be noted, however, that the large majority (89%) of simulated values at

individual stations in China are within a factor 2 of the measurements. Figure S1 shows the model is also able to capture daily variability in measured $PM_{2.5}$ concentrations at three Chinese megacities; simulating daily mean concentrations within a factor 1.8 of the measurements (NMBF=0.09-0.80; r=0.47-0.56).

The model is expected to underestimate measured concentrations in countries located towards the boundaries of the regional model domain (the Philippines, Pakistan and Republic of Korea) due to increased influence from the coarse resolution global

model and potential missing sources outside the regional model domain. Therefore, we do not present results for these countries in the following sections.



### 3.2 Contribution of emission sectors to ambient PM$_{2.5}$ concentrations

### 3.2.1 Contribution of emission sectors to PM$_{2.5}$ by country

Figure 2 shows the percentage contribution of each anthropogenic emission sector to the simulated population-weighted annual mean PM$_{2.5}$ concentration for each country within the model domain. The relative contribution of each sector is calculated for

each country as the percentage difference between the simulated population-weighted annual mean PM$_{2.5}$ concentrations from the control simulation (with all sources included) and from each of the individual eliminated-sector simulations. Results for Afghanistan, Pakistan, the Philippines and South Korea are not shown in Fig. 2 due to their proximity to the edges of the model domain (Sect. 3.1).

In China, the largest contributions to population-weighted annual mean PM$_{2.5}$ concentrations are from the industrial (43%) and

residential (38%) emission sectors, which is consistent with previous studies (see Sect. 4). The next largest contributions are from natural and minor sources (including mineral dust, sea spray and biogenic SOA) (9%), power generation (5%) and road transport (4%). In India, the population-weighted annual mean PM$_{2.5}$ is dominated by the contribution from the residential sector (52%) as reported in Conibear et al. (2018a), with power generation, industry and transport contributing 21%, 16% and 10%, respectively. Open biomass burning emissions contribute relatively small fractions to the population-weighted annual

mean PM$_{2.5}$ in both China (1%) and India (3%). However, it is likely that fire emission datasets underestimate the emissions from agricultural fires in China (e.g. Zhang et al., 2016) and India (e.g. Cusworth et al., 2018).

In India, there is a noticeably larger fractional contribution of power generation emissions to the population-weighted annual mean PM$_{2.5}$ concentration (21%) compared with China (5%). This is likely due to multiple reasons including lack of regulation, lack of flue-gas desulphurisation, and low energy efficiencies in India (Venkataraman et al., 2018), resulting in higher implied

emission factors (emissions per unit of activity) for PM$_{2.5}$ from power generation in India relative to China (Janssens-Maenhout et al., 2015) and higher fractional contributions of power generation to total primary PM$_{2.5}$ emissions (16% of total in India; 7% in China (Li et al., 2017)). Conversely there is a larger contribution of industrial emissions to population-weighted annual mean PM$_{2.5}$ concentration in China (43%) than in India (16%). This is likely due to a larger amount of heavy industry in China compared to in India (primary PM$_{2.5}$ emissions from industry contribute 50% to the total emitted PM$_{2.5}$ in China compared to

18% in India (Li et al., 2017)). This is likely to change in the future in India, where industry becomes dominant under current policies (Conibear et al. (2018b)).

In Bangladesh, the contributions to population-weighted annual mean PM$_{2.5}$ are very similar to those in India, with a larger contribution from the residential sector (58%) and slightly smaller contributions from power generation (17%) and transport (7%) emissions. The contributions from industry (16%) and open biomass burning (3%) match those in India. In Nepal and

Bhutan, residential emissions are even more dominant, contributing 67-68% of population-weighted annual mean PM$_{2.5}$.

The residential sector also dominates contributions to population-weighted annual PM$_{2.5}$ in Myanmar (38%), Vietnam (52%) and Cambodia (45%). Industrial emissions contribute the largest fraction of population-weighted PM$_{2.5}$ in Thailand (34%),





with relatively large contributions in Laos (19%) and Vietnam (23%). In Laos, the population-weighted $PM_{2.5}$ is dominated by emissions from open biomass burning (30%). It is likely that open biomass burning emissions are underestimated in Southeast Asia (Reddington et al., 2016; Lasko et al., 2017), and so may make a larger contribution to $PM_{2.5}$ concentrations than reported here.

The contribution of natural sources (e.g. biogenic SOA, sea spray and mineral dust) and minor sources to population-weighted annual mean $PM_{2.5}$ is relatively large in China and Mainland Southeast Asia compared to the Indian Subcontinent. Shi et al. (2016) also found a relatively large combined contribution from windblown dust, SOA and sea salt to province-average $PM_{2.5}$ concentrations in China (17%; calculated as the average over the provinces included in our model domain).

The residual $PM_{2.5}$ concentration classed as from "natural and minor" sources also depends on the non-linear effects of
simulated air pollutant concentrations when emissions are eliminated in the model. Since the atmospheric chemistry, aerosol processes and meteorology are fully coupled in WRF-Chem, eliminating primary air pollutant emissions may act to increase $PM_{2.5}$ concentrations through changes in wind speed, boundary layer depth, secondary aerosol formation, aerosol removal etc. This would act to increase the calculated contribution of "natural and minor" sources to simulated population-weighted annual mean $PM_{2.5}$ concentrations, although this is typically less than 1%.

**3.2.2 Contribution of emission sectors to $PM_{2.5}$ by state or province**

Figure 3 shows the contribution of each emission sector to the population-weighted annual mean $PM_{2.5}$ concentration in each province in China (within the model domain) and each state in India. In all Chinese provinces, either industrial or residential emissions make the largest contributions to population-weighted annual mean $PM_{2.5}$ concentrations, with the exception of Hainan Island where natural and minor sources make the largest contribution (Fig. 3a). The contributions from residential
emissions range from 17 to 50%, in general with larger contributions from this sector in northern, western and central provinces compared to southern and south-eastern provinces e.g. contributions in Beijing (41%), Sichuan (49%) and Hubei (41%) compared to Guangdong (26%) and Shanghai (17%). This is due to greater emissions from heating in colder northern and mountainous regions in winter months (Archer-Nicholls et al., 2016). The contribution of the industrial sector to population-weighted annual mean $PM_{2.5}$ is prevalent across all provinces (range 23 to 60%), with the largest contributions in the major
steel-producing provinces of Hebei (47%) and Jiangsu (47%), in the major coal-producing province of Shanxi (52%) and in Shanghai (60%).

The contributions from the other emission sectors (land transport, power generation, agriculture, shipping and open biomass burning) to population-weighted annual mean $PM_{2.5}$ are relatively small (<13%) in all provinces. The contribution of power generation emissions ranges from 3% to 11%, with the greatest contribution in the provinces of Zhejiang (9%) and Ningxia
(11%). The land transport sector generally makes the largest contributions in eastern and south-eastern provinces relative to provinces in other regions of China, with largest the contributions in Shanghai (6%) and Beijng (6%). We find that the





contributions of shipping and agricultural emissions across China are particularly small relative to the other sectors, with the largest contributions in the Special Administrative Region (SAR) of Hong Kong (2.5% and 0.5%, respectively).

The largest contributions from open biomass burning emissions are seen in the south-western and southern provinces of China, with the largest contribution in Yunnan province (12%). These provinces are influenced by transport of smoke from fires in Mainland Southeast Asia and northeast India during the burning season (~February to April; see Fig. 5) (Huang et al., 2013; Zhu et al., 2017). Local fires also occur in these regions (Zhang et al., 2016; Zhu et al., 2017; Zhou et al., 2017) which will also contribute to simulated province-average $PM_{2.5}$ concentrations.

In India (Fig, 3b), residential emissions make the largest contribution to population-weighted annual mean $PM_{2.5}$ concentrations in all states (range 29 to 64%), with the exception of Delhi, where road transport contributes the largest fraction (as reported by Conibear et al. (2018a)). In general, the contributions of residential emissions are larger than in Chinese provinces, particularly in the northern and northeastern states, with the largest contributions in West Bengal (61%), Sikkim (60%), Assam (60%), and Bihar (64%). Land transport emissions also generally contribute a larger fraction to the population-weighted annual mean $PM_{2.5}$ in Indian states (range 6 to 34%) compared to in Chinese provinces (range 1 to 6%), with the largest contributions in Delhi (34%) and Haryana (25%).

The power generation sector makes relatively large contributions to the population-weighted annual mean $PM_{2.5}$ across India (range 13 to 31%), with larger contributions in all Indian states compared to Chinese provinces within the model domain (range 3 to 10%). The largest contributions of power generation emissions are in the states of Central India: Chhattisgarh (31%), Jharkhand (25%), Maharashtra (24%) and Andhra Pradesh (25%), likely due to the large coal-fired power plants located in these states (clustered at the pit heads of coal mines; Guttikunda and Jawahar (2014)). In contrast, contributions from the industrial sector are smaller in almost all states in India (range 11 to 26%) compared to the provinces in China (range 23 to 60%), with the largest contributions in Gujarat (26%) and Maharashtra (20%).

Open biomass burning emissions make relatively large contributions to $PM_{2.5}$ in northern and northeastern states in India, particularly in Mizoram (27%), Manipur (23%) and Nagaland (22%). Agricultural fires (involving burning of crop residues) are widespread across northern India (Vadrevu et al., 2015) with substantial impacts on regional air quality (Liu et al., 2018; Sakar et al., 2018). Northeastern states may also be affected by transported smoke from deforestation and agricultural fires in neighbouring Myanmar.

### 3.2.3 Dominant emission sector contributions to $PM_{2.5}$

Figure 4 shows the spatial distribution of the anthropogenic emission sectors that yield the largest reduction in simulated annual mean surface $PM_{2.5}$ concentrations. Over the majority of the Indian Subcontinent, excluding residential emissions leads to the largest reduction annual mean $PM_{2.5}$. In some small regions of India, the largest reductions in $PM_{2.5}$ are achieved by excluding





the power generation (in parts of central-east India), transport (in Delhi), and industrial (in eastern Maharashtra and central Gujarat) sectors.

Excluding residential emissions also yields the largest reductions in annual mean $PM_{2.5}$, relative to the other emission sectors, in Vietnam, southern Myanmar, central Laos and Cambodia, and southern and eastern parts of China. In central and south-eastern China and central Thailand, the largest reductions in annual mean $PM_{2.5}$ are achieved by excluding industrial emissions. In other parts of Mainland Southeast Asia (northern and eastern regions of Myanmar and Thailand, and northern and southern regions of Cambodia and Laos), excluding fire emissions gives the largest reductions in simulated annual mean $PM_{2.5}$ concentrations relative to the other emission sectors.

### 3.2.4 Seasonal variation in dominant emission sector contributions to $PM_{2.5}$

Figure 5 shows the seasonal variation in the dominant emission sectors contributing to surface $PM_{2.5}$ over the South Asia and East Asia model domains. Seasonal variation in anthropogenic sources contributing to $PM_{2.5}$ is relatively low over much of the Indian Subcontinent. Over this region, excluding emissions from residential energy use yields the largest reduction in seasonal mean $PM_{2.5}$ concentrations throughout the year, with a small increase in the areas dominated by industrial emissions (in Maharashtra and Gujarat in western India) during March to August and power generation emissions (in central India) during March to May. In northeastern India, the dominant emission sector switches from residential to open biomass burning during March to May. Open biomass burning emissions can also be seen to dominate over residential emissions in northern India (states of Punjab and Haryana) during September to November, likely due to agricultural burning of rice residues.

In contrast to India, there is strong seasonal variation in the dominant emission sectors in Mainland Southeast Asia. During December to February, excluding emissions from residential energy use yields the largest reduction in seasonal mean $PM_{2.5}$ over much of the region, with fire emissions dominating seasonal mean $PM_{2.5}$ in Cambodia. During March to May, excluding fire emissions yields the largest reduction in seasonal mean $PM_{2.5}$ over most of Mainland Southeast Asia, but also in Taiwan, northern Philippines, eastern India, and south-west China. During July to November, the largest reductions in seasonal mean $PM_{2.5}$ are achieved by excluding industrial emissions in central and southern Thailand (and Laos during September to November), power generation emissions in northern Thailand and residential emissions in Myanmar, Cambodia and Vietnam.

In China, excluding emissions from residential energy use yields the largest reduction in seasonal mean $PM_{2.5}$ concentrations during the winter months (December to February), with the exception of the heavily industrialised regions of the Pearl River Delta (PRD) and Yangtze River Delta (YRD) where industrial emissions dominate. During March to November, excluding either residential or industrial emissions yield the largest reductions in seasonal mean $PM_{2.5}$ in central, eastern and south-eastern China, depending on the specific region.



### 3.3 Impacts of emission sectors on human health

Table 1 shows the percentage of population exposed to $PM_{2.5}$ concentrations above the WHO Air Quality Guideline (AQG) limits for each country in the model domain. Our model simulations show that in 2014, the vast majority of the South and East Asian population was exposed to annual mean $PM_{2.5}$ concentrations in excess of the WHO AQG of 10 µg m$^{-3}$ (range per country: 43-100%) and the WHO Level 2 Interim Target (IT-2) of 25 µg m$^{-3}$ (range per country: 0-100%).

Figure 6a shows the total annual premature mortality due to long-term exposure to ambient $PM_{2.5}$ from all sources in India, China, and countries in Mainland Southeast Asia. The spatial distribution of $PM_{2.5}$-related disease burden in South and East Asia is shown in Fig. S2. We estimate the total annual premature mortality in China (including Taiwan) to be 1,047,000 (95% uncertainty interval (95UI): 846,000–1,287,000), with 19,679,000 (95UI: 15,622,000–24,580,000) years of life lost (YLL) compared to 990,000 (95UI: 660,000–1,350,000) annual premature mortalities and 24,606,000 (95UI: 14,567,000–32,698,000) YLL in India (Conibear et al., 2018a). The disease burden attributable to exposure to ambient $PM_{2.5}$ in China is dominated by stroke (29%; Fig. 6a) IHD (26%) and COPD (26%), with smaller contributions from LC (13%) and LRI (6%). In India, the fractions of mortality attributable to stroke (14%) and LC (2%) are less than in China, with larger fractions from COPD (31%), IHD (35%) and LRI (17%).

In Mainland Southeast Asia, we estimate the total annual premature mortality as 109,000 (95UI: 66,000–160,000) with 2,304,000 (95UI: 1,309,000–3,540,000) YLL. The fraction of premature mortality estimated for each country in Southeast Asia scales roughly with population, with the largest fractions in Vietnam (42%) and Thailand (31%) and smallest in Laos (3%). The disease burden is dominated by IHD in Cambodia (40%) and Laos (37%), by stroke in Vietnam (33%) and Myanmar (33%), and by LRI in Thailand (31%).

Our estimates of the total premature mortality due to long-term exposure to ambient $PM_{2.5}$ compare well with those from GBD2015 (Cohen et al., 2017) for China, India and countries in Southeast Asia and (Fig. S3a). The mean estimates from this study lie well within the uncertainty bounds of the values reported by Cohen et al. (2017) for each country, with the exception of Myanmar. For Myanmar, the mean value of Cohen et al. (2017) is higher than the value from this study by a factor 1.5, but lies within our estimated uncertainty range.

Figure 6b and Table 2 show the sector-specific averted annual premature mortality due to a reduction in exposure to ambient $PM_{2.5}$, using the "substitution" method as described in Sect 2.2 and Conibear et al. (2018a). The spatial distribution of averted disease burden is shown in Fig. S2b-h. The summation of sector contributions is 437,000 (95UI: 327,000–583,000) premature mortalities per year in China and Taiwan (42% of the control simulation), 48,000 (95UI: 27,000–74,000) premature mortalities per year in Southeast Asia (44% of the control simulation) and 469,000 (95UI: 304,000–626,000) premature mortalities per year in India (47% of the control simulation; Conibear et al., 2018a). It is important to note that these values are substantially lower than if we were to use the attribution method as used in other studies (e.g. Lelieveld et al., 2015; Archer-Nicholls et al., 2016; GBD-MAPS, 2016; Gao et al., 2018) because of the non-linear exposure-response relationship (Conibear et al., 2018a).





When using the attribution method, Conibear et al. (2018a) obtained a summation of 1,012,000 (95UI: 675,000–1,381,000) annual premature mortalities in India; equivalent to 102% of the control simulation.

The industrial emission sector is the dominant contributor to premature mortalities due to exposure to ambient $PM_{2.5}$ in China and Thailand. Eliminating emissions from the industrial emission sector would avert 204,000 (95UI: 152,000–271,000) annual premature mortalities in China and 13,000 (8,000–20,000) annual premature mortalities across Southeast Asia.

Residential energy use is the dominant contributor to premature mortalities due to exposure to ambient $PM_{2.5}$ in Vietnam, Myanmar and Cambodia and the second largest contributor in China, Thailand and Laos. Eliminating emissions from residential energy use would avert 188,000 (95UI: 141,000–250,000) and 24,000 (95UI: 13,000–36,000) annual premature mortalities in China and Southeast Asia, respectively.

Open biomass burning is the dominant contributor to premature mortalities due to exposure to ambient $PM_{2.5}$ in Laos. Preventing open biomass burning in East Asia would avert 8,000 (95UI: 4,000-13,000) annual premature mortalities across Southeast Asia and 7,000 (95UI: 6,000-9,000) annual premature mortalities in China.

The land transport and energy generation emission sectors are not dominant contributors to the national/regional annual premature mortality estimates in Fig. 6 and Table 2. However, eliminating emissions from these sectors would still yield a substantial human health benefit in China, averting 15,000 (95UI: 11,000-20,000) and 22,300 (95UI:16,000-30,000) annual premature mortalities, respectively.

## 4. Comparison to previous studies

Table 3 summarises the previous studies that have quantified the emission source/sector contributions to $PM_{2.5}$ and associated health burden in China and India. These studies have used a range of different of approaches, methods and tools, which lead to a wide range in estimates of sector-specific contributions to $PM_{2.5}$ concentrations (Fig. 7; Tables 4 and 5) and annual premature mortalities (Figs. 8 and S3; Tables S1 and S2).

For China we compare the total annual premature mortality estimate from this study to estimates from the previous studies listed in Table 3 (Fig. S3b). Our estimate (1,046,900 (95UI: 846,100- 1,286,900)) sits well within the multi-model range of 916,000 to 1,357,000 (UI: 594,000–1,915,000) premature mortalities. Despite the large differences in modelling tools, emissions inventories and health functions used in these studies, our estimate (and uncertainty range) for China overlaps with all previous estimates in Fig. S3 apart from Lelieveld et al. (2015) (whose estimate also includes premature mortality due to exposure to ozone and does not report a UI specifically for China). We note that the larger mortality estimate from Lelieveld et al. (2015) will primarily be due to the GBD2010 exposure-response function, which predicted much larger relative risks for cardiovascular diseases (IHD and stroke) compared to relative risks from GBD2015. The multi-model mean for China is: 1,135,000 (UI: 746,000-1,398,000) annual premature mortalities. It is important to note that these estimates apply to a range



of years (ranging from 2001 to 2014 in terms of meteorology and from 2005 to 2015 in terms of anthropogenic emissions; Table 3).

Figure 7 compares estimates of sector-specific contributions to annual mean $PM_{2.5}$ concentrations in China and India. Previous studies consistently find that residential energy use and industry are the dominant emission sectors in China for annual mean

$PM_{2.5}$ (Fig. 7a and Table 4). Residential emissions contribute an average of 26% (13-38%) and industrial emissions contribute an average of 30% (8-43%) to annual mean $PM_{2.5}$ concentrations in China (see Fig. 7a and Table 4). Other sectors make a smaller contribution, with emissions from power generation contributing an average of 14% (range 5-33%), land transport an average of 7% (range 3-15%), open biomass burning an average of 4% (range 1-8%) and agriculture an average of 13% (range 0.1-29%).

In India, previous studies consistently find that residential emissions dominate contributions to annual mean $PM_{2.5}$ concentrations (Fig. 7b and Table 5), with an average contribution of 38% (22-56%) over all studies. Other sectors make a smaller contribution, with emissions from industry contributing an average of 14% (range 7-20%), power generation an average of 18% (range 7-40%), land transport an average of 8% (range 2-20%), open biomass burning an average of 5% (range 3-7%) and agriculture an average of 6% (range 0.3-12%).

Although previous studies consistently agree on the dominant emission sectors contributing to ambient $PM_{2.5}$ concentrations in India and China, there is considerable variability in the estimated contribution from each sector. For most sectors the fractional contribution from any one sector varies by a factor of 2 to 5, with the largest range for open biomass burning (up to a factor of 8) and agriculture (greater than a factor of 10). There have been fewer studies quantifying the contribution of agriculture to $PM_{2.5}$ concentrations in China and India, and the contribution of this sector has the largest uncertainty. Following

this, our study is the only one in Table 3 to quantify the contribution of shipping emissions to population-weighted annual mean $PM_{2.5}$, and so the contribution of this sector is also likely to be uncertain. However, we notes that the contribution of shipping emissions to $PM_{2.5}$ concentrations is only likely to be important for coastal regions (Lv et al., 2018) and relatively small compared to other emission sectors.

The different model simulation and anthropogenic emission years will contribute to the range across previous studies,

particularly since China and India have experienced rapid changes in emissions in the last decade (Saikawa et al., 2017; Zheng et al., 2018). Reducing the multi-model range in the future will require up-to-date and consistent anthropogenic emissions inventories (with improved quantification of the fractional contributions of the different sectors) to use in air quality models. It will also be important to run the same air quality models at different spatial resolutions to ensure that the fractional contributions of some sectors (e.g. land transport and residential energy use) to ambient $PM_{2.5}$ concentrations are not

underestimated due to missing or underrepresented sub-grid emission sources. Model grid resolution is also important to consider when estimating the health impacts of emissions from different sectors, particularly for land transport and residential energy use, where the exposure (or intake fraction) depends strongly on co-location of sources and high population (U.S.





National Research Council, 2012). Comparing model results of emission sector contributions with in-situ, source apportionment measurements (as in Karagulian et al. (2017)) may help to constrain the range in multi-model estimates.

The large variability in the disease burden estimates (Tables S1 and S2) are strongly influenced by the exposure-response function used in each study. The IER functions were developed for GBD2010 by Burnett et al., (2014). Each subsequent GBD

study (2013, 2015, 2016, and 2017) updates the coefficients used to calculate relative risk within the IER functions (Sects. 2.2 and S1) due to the incorporation of more epidemiological evidence. In general, with the same $PM_{2.5}$ concentration fields, applying coefficients from GBD2010 will yield the highest estimates of relative risk and mortality; applying coefficients from GBD2013 will yield the lowest estimates; while applying coefficients from GBD2015 and GBD2016 will yield medium estimates. Results from GBD2017 give slightly lower estimates of risk and mortality than GBD2015 and GBD2016, primarily

due to the different approach to combine risk from household and ambient $PM_{2.5}$ and avoid overestimation for those exposed to both. A recent study that constructed a $PM_{2.5}$-mortality hazard ratio function based only on cohort studies of ambient air pollution, rather than the IER approach of integrating several sources (ambient and household air pollution, passive and active smoking), finds estimates that are 120% higher than the GBD2015 IER (Burnett et al., 2018). Future work should move to using consistent and up-to-date exposure-response functions to reduce the multi-model range in health impact estimates,

although the associated uncertainty range will likely remain large.

## 5. Summary and conclusions

In this study we used a high-resolution air quality model to explore the contribution of seven different anthropogenic emission sectors to surface $PM_{2.5}$ concentrations across South and East Asia, and calculated the human health impacts if emissions from each of these sectors were to be eliminated.

We found that the vast majority of the South and East Asian populations are exposed to annual mean $PM_{2.5}$ concentrations exceeding the WHO Air Quality Guideline, which we estimated to cause 1,047,000 (95U: 846,000–1,287,000), 990,000 (95UI: 660,000–1,350,000), and 109,000 (95UI: 66,000–160,000) annual premature mortalities in China, India and Mainland Southeast Asia, respectively. Emissions from the residential, industrial and open biomass burning sectors dominate contributions to population-weighted annual mean $PM_{2.5}$ concentrations in South and East Asia. Eliminating emissions from

these sources would substantially reduce the population exposed to ambient concentrations of $PM_{2.5}$ above the WHO Air Quality Guideline and avert numerous $PM_{2.5}$-related premature mortalities and years of life lost.

In China, we found that eliminating emissions from the industrial sector yielded the largest reduction in population-weighted annual mean $PM_{2.5}$ concentrations (by 43% in our study; on average 29% across previous studies); averting the largest number of annual premature mortalities (204,000 (95UI: 152,000-271,000) in our study). Eliminating residential solid-fuel combustion

also yielded substantial reductions in population-weighted annual mean $PM_{2.5}$ concentrations (by 38% in our study, on average 29% across previous studies) and annual $PM_{2.5}$-related premature mortalities (188,000 (95UI: 141,000–250,000) in our study).



In Southeast Asia, eliminating emissions from residential solid-fuel combustion yielded the largest reductions in population-weighted annual $PM_{2.5}$ in Myanmar (by 38%), Vietnam (by 52%) and Cambodia (by 45%) and the second largest reductions in Thailand (by 20%) and Laos (by 25%). Removing this sector would avert 24,000 (95UI: 13,000-36,000) annual premature mortalities across the region. Other important emission sectors in this region are industry and open biomass burning, removing

these emissions would avert 13,000 (95UI: 8,000-20,000) and 8,000 (95UI: 4,000-13,000) annual premature mortalities in Southeast Asia, respectively.

Future work should focus on identifying the most effective options within the residential, industrial and open biomass burning emission sectors to improve air quality across South and East Asia. For the residential sector, switching from solid-fuel combustion to combustion of clean fuels (such as Liquefied Petroleum Gas (LPG)) will likely be the most effective option.

Large reductions in ambient $PM_{2.5}$ concentrations have already been achieved in China between 2005 and 2015, which may have been driven by a reduction in residential emissions from widespread adoption of clean fuels (due to increasing wealth and urbanisation rather than control policies) (Zhao et al., 2018). However, despite reductions in ambient $PM_{2.5}$ concentrations, exposure to air pollution in China remains a leading risk factor for human health. In India, there are programmes now in place to promote LPG to the poorest households (Goldemberg et al., 2018), aiming to increase the use of LPG from 30% in 2015 to

90% by the early 2020's. The air quality benefits of these programmes in India are yet to be explored.

Anthropogenic emissions are changing rapidly across Asia, leading to large changes in air pollutant concentrations (e.g. Silver et al., 2018), so future work should include more up-to-date emission inventories that are becoming available for China and India to explore how the contributions of emission sectors to $PM_{2.5}$ pollution have changed over time. There is a strong need for development of up-to-date anthropogenic emission inventories for countries in Southeast Asia to improve our

understanding of the contributions of pollution sources in this region for recent years.

Previous studies agree that emissions from the residential and industrial sectors dominate population-weighted $PM_{2.5}$ concentrations in China and emissions from the residential sector dominate in India. Despite this qualitative agreement, we found the contribution of individual sectors varied by a factor of 2-5 or more. It will be important for future work to explore the reasons for these differences between model estimates of the contribution of different sources to air pollutant concentrations

and the associated health burden.

This study can inform effective emission-reduction strategies at the local level across South and East Asia to improve air quality and reduce the substantial disease burden from air pollution exposure. Our work has demonstrated that the combustion of solid fuels dominates contributions to ambient $PM_{2.5}$ concentrations and associated health effects in India, China and Mainland Southeast Asia. We therefore recommend that emission-reduction strategies in these countries should focus on

reducing the combustion of solid fuels in homes, industry, and through open burning.



**Author contributions**

C.L.R. performed the model simulations and analysed the model data. L.C. performed the health impact calculations. All authors contributed to scientific discussions and helped write the manuscript.

**Acknowledgements**

We acknowledge the AIA Group and Natural Environment Research Council (NE/N006895/1) for funding. This work made use of the facilities of N8 HPC Centre of Excellence, provided and funded by the N8 consortium and EPSRC (EP/K000225/1). We acknowledge use of the WRF-Chem preprocessor tool bio_emiss, fire_emiss, and mozbc provided by the Atmospheric Chemistry Observations and Modeling Lab (ACOM) of NCAR. We acknowledge use of NCAR/ACOM MOZART-4 global model output available at http://www.acom.ucar.edu/wrf-chem/mozart.shtml.

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



**Table 1.** Summary of annual mean $PM_{2.5}$ measurements from the World Health Organization (WHO) Ambient (outdoor) air quality database (2016, 2018). The table shows the number of stations with available data, the year(s) the measurements were conducted and the number of reported $PM_{2.5}$ concentrations that were converted from $PM_{10}$ measurements (see Sect. 2.3). The model normalised mean bias factor (NMBF; Yu et al., 2006) and Pearson's correlation coefficient (r) against observations are given for each country with available WHO measurements. The simulated population-weighted annual mean $PM_{2.5}$ concentration is given for each country within the model domain (shown in Fig 1) and the percentage of population "exposed to" (in the same model grid cell as) annual mean $PM_{2.5}$ concentrations greater than the WHO Air Quality Guideline (AQG; 10 $\mu$g m$^{-3}$) and WHO Interim Target 2 (IT-2; 25 $\mu$g m$^{-3}$) (WHO, 2006; 2016).

| Country | No. of stations | Year(s) of measurements | Measured/ converted $PM_{2.5}$ | Model NMBF; r | Model population-weighted $PM_{2.5}$ ($\mu$g m$^{-3}$) | % of population exposed to $PM_{2.5}$ > WHO AQG; WHO IT-2 |
|---|---|---|---|---|---|---|
| Bangladesh | 8 | 2014 | Measured | -0.26; 0.33 | 67.1 | 100%; 100% |
| Bhutan | 4 | 2013, 2014 | Converted | -0.63; 0.41 | 46.3 | 100%; 92% |
| Cambodia | - | - | - | - | 24.4 | 100%; 40% |
| China | 193 | 2014 | Measured: 192 Converted 1 | +0.33; 0.76 | 72.3 | 97%; 94% |
| India | 127 | 2012-2016 | Measured: 21 Converted: 106 | -0.05; 0.37 | 57.7 | 99%; 97% |
| Rep. of Korea | 15 | 2014 | Converted | -0.32; 0.11 | 20.4 | 98%; 16% |
| Laos | - | - | - | - | 27.2 | 100%; 72% |
| Myanmar | 16 | 2009, 2012, 2013, 2015 | Converted | -1.27; 0.34 | 25.7 | 100%; 60% |
| Nepal | 1 | 2013 | Measured | -0.81; - | 50.6 | 100%; 88% |
| Pakistan | 6 | 2009-2011, 2013 | Measured | -0.80; 0.64 | 38.8 | 96%; 65% |
| Philippines | 19 | 2013, 2015, 2016 | Measured: 14 Converted: 5 | -1.05; 0.19 | 8.1 | 43%; 0% |
| Thailand | 22 | 2014 | Converted | +0.06; 0.38 | 24.5 | 89%; 57% |
| Vietnam | 2 | 2016 | Measured: 2 | +0.46; - | 44.2 | 100%; 81% |





**Table 2.** Estimated total annual premature mortality due to exposure to ambient PM$_{2.5}$ in countries in South and East Asia. Also shown is the averted annual premature mortality per country due to a reduction in exposure to ambient PM$_{2.5}$, calculated using the substitution method. Averted premature mortality estimates are given for each emission sector: agriculture (AGR), biomass burning (BBU), power generation (ENE), industrial non-power (IND), residential energy use (RES), shipping (SHP; East Asia only) and land transport (TRA). Values in bold show the emission sector that gives the largest averted premature mortality for each country/region. "SE Asia" includes Myanmar, Thailand, Laos, Cambodia and Vietnam (results for these countries are also shown separately). China includes Hong Kong SAR, Macau SAR and Taiwan. Values in parentheses represent the 95% uncertainty intervals (95UI). Values are rounded to the nearest 100. Negative values represent increases in estmiated premature mortality when an emission sector is removed (due to a increase in simulated PM$_{25}$ concentrations).

| Country/ region | All sources | AGR | BBU | ENE | IND | RES | SHP | TRA |
|---|---|---|---|---|---|---|---|---|
| China (incl. Taiwan) | 1,046,900 (846,100-1,286,900) | 500 (300-700) | 7,300 (5,600-9,300) | 22,300 (16,500-30,400) | **203,600 (152,300-271,100)** | 187,900 (140,700-250,300) | 700 (500-900) | 14,800 (10,800-20,500) |
| India | 990,000 (660,200-1,350,800) | 1,000 (700-1,400) | 12,300 (8,400-16,450) | 90,400 (59,600-121,500) | 66,500 (44,700-89,600) | **255,600 (161,800-339,700)** | - | 43,000 (28,900-57,900) |
| SE Asia | 108,700 (65,800-160,000) | -200 (-300 to -100) | 8,200 (4,400-12,800) | 1,900 (1,100-3,000) | 13,300 (7,600-20,000) | **23,700 (13,200-36,200)** | 100 (100-100) | 1,200 (700-1,900) |
| Myanmar | 20,200 (10,100-33,100) | 0 (-100-0) | 3,000 (1,400-5,200) | 400 (200-600) | 1,300 (600-2,200) | **4,800 (2,200-8,000)** | 0 (-100-0) | 100 (100-300) |
| Thailand | 33,400 (21,100-47,800) | -100 (-100-0) | 3,100 (1,800-4,600) | 900 (500-1,300) | **6,600 (3,900-9,700)** | 4,000 (2,400-5,900) | 0 (0-0) | 700 (400-1,100) |
| Laos | 3,000 (1,800-4,500) | 0 (0-0) | **500 (300-800)** | 100 (0-100) | 300 (200-400) | 400 (200-700) | 0 (0-0) | 0 (0-0) |
| Cambodia | 6,500 (4,100-9,200) | 0 (0-0) | 500 (300-700) | 100 (0-100) | 400 (200-600) | **1,700 (1,000-2,500)** | 0 (0-0) | 100 (0-100) |
| Vietnam | 45,600 (28,500-65,400) | -100 (-100-0) | 1,000 (600-1,500) | 500 (300-800) | 4,700 (2,700-7,100) | **12,800 (7,400-19,000)** | 100 (100-200) | 300 (200-400) |



**Table 3.** Summary of studies quanitfying sector-specific contributions to PM<sub>2.5</sub> concentrations and PM<sub>2.5</sub>-related disease burden in China and/or India (shown in order of publication year). The approaches used to estimate the sector-specific contributions are given based on desriptions included in the published papers and suplementary information. The model grid spacing/resoluton is given in terms of longitude x latitiude (for grid reolsutions in degrees, approximate conversions to km at the equator are also given).

| Reference | Estimation approach for sector-specific contributions | | Meteorology year | Region | Model (grid resolution) | Anthropogenic emissions | Exposure-response function |
| | PM$_{2.5}$ | Health burden | | | | | |
|---|---|---|---|---|---|---|---|
| Lelieveld et al. (2015) | Source-removal | Attribution | 2010 | Global | EMAC (1.1°×1.1° ~122 x 122 km) | EDGAR for 2010 | GBD2010 |
| Silva et al. (2016) | Source-removal | Substitution | 2005 | Global | MOZART-4 (0.50°x0.67° ~56 x 74 km) | RCP8.5 for 2005[a] | GBD2010 |
| Archer-Nicholls et al. (2016) | Source-removal | Attribution | 2014 | China | WRF-Chem v3.6.1 (27 x 27 km) | EDGAR-HTAPv2 for 2010 | GBD2013 |
| GBD-MAPS (2016) | Source-tagging | Attribution | 2012 | China | GEOS-Chem East Asia (0.50° x0.67° ~56 x 74 km)[b] | MIX (for 2010) updated for 2013 | GBD2013 |
| Butt et al. (2016) | Source-removal | Substitution | 2000 | Global | GLOMAP (2.8° x2.8° ~310 x 312 km) | MACCity for 2000 | Ostro (2004) |
| Karagulian et al. (2017) | Source-tagging | N/A | 2001 | Global | TM5-FASST (1°×1° ~110 x 110 km) | EDGAR-HTAP2 for 2010 | N/A |
| Shi et al. (2017) | Source-tagging | N/A | 2013 | China | Source-oriented CMAQ (36 x 36 km) | MEIC for 2013 | N/A |
| Hu et al. (2017) | Source-tagging | Attribution | Not specified | China | WRF v3.6.1 + Source oriented CMAQ (36 x 36 km) | MEIC for 2013 | GBD2010 |
| Aunan et al. (2018) | Source-removal | Substitution | 2012 | China | GEOS-Chem East Asia (0.50° x0.67° ~56 x 74 km) | 2010 emissions updated for 2013 (Ma et al., 2017) | GBD2010 (Lookup table from Apte et al. (2015)[c]) |
| Gao et al. (2018) | Source-tagging | Attribution | 2013 | China & India | WRF-Chem v3.6.1 (60 x 60 km) | MIX for 2010 (with MEIC for 2013) | GBD2015 |
| GBD-MAPS (2018) | Source-removal | Attribution | 2012 | India | GEOS-Chem South Asia (0.50°x0.67° ~56 x 74 km)[b] | IITB for 2015[d] | GBD2015 |
| Gu et al. (2018) | Source-removal | Attribution | 2010 | China | WRF v3.7.1 + CMAQ v4.7.1 (27 x 27 km) | HTAPv2 for 2010 | Gu and Yim (2016) |



| Guo et al. (2018) | Source-tagging | Attribution | Not specified | India | WRF v3.7.1 + CMAQ 5.0.1 (36 x 36 km) | EDGAR v4.3.1 for 2010 | GBD2010 |
|---|---|---|---|---|---|---|---|
| Upadhyay et al. (2018) | Source-removal | Substitution | 2010 | India | WRF-Chem v3.6 (10 x 10 km) | EDGAR-HTAPv2 for 2010 | GBD2015; Chowdhury and Dey, 2016 |
| Butt et al., *in prep.* (2019) | Source-removal | Substitution | 2015 | Global | TOMCAT-GLOMAP (2.8° x2.8° ~310 x 312 km) | ECLIPSE for 2015 | GBD2015 |
| This study & Conibear et al. (2018a) | Source-removal | Substitution & attribution | 2014 | South & East Asia | WRF-Chem v3.7.1 (30 x 30 km) | EDGAR-HTAPv2 for 2010 | GBD2015 |

[a] Representative Concentration Pathway 8.5 global emissions inventory for 2005 (Riahi et al. 2011).
[b] Spatially resolved fractional contributions of different source sectors estimated with GEOS-Chem simulations were multiplied by high-resolution ambient $PM_{2.5}$ concentration estimates developed for the GBD2015 project to estimate the ambient $PM_{2.5}$ concentrations attributable to each source sector.
5  [c] Derived from the IER functions for exposure to $PM_{2.5}$ and five mortality end-points, as established by Burnett et al. (2014).
[d] IITB (the India Institute of Technology – Bombay) emission inventory (see GBD-MAPS (2018)).



**Table 4.** Comparison of relative sector-specific contributions to simulated annual mean PM$_{2.5}$ concentrations over China from this study and previous studies. Emission sectors are: agriculture (AGR), biomass burning (BBU), power generation (ENE), industrial non-power (IND), residential energy use (RES), land transport (TRA) and shipping (SHP). The largest relative contribution for each study is in bold. The average over all studies (multi-model mean) is shown for population-weighted, area-weighted, and all annual mean PM$_{2.5}$ concentrations and relative contributions.

| Reference | Population-weighted or area-weighted annual mean PM$_{2.5}$ | Annual mean PM$_{2.5}$ concentration for China | Relative sector-specific contributions to simulated annual mean PM$_{2.5}$ concentrations (%) | | | | | | |
|---|---|---|---|---|---|---|---|---|---|
| | | | RES | IND | ENE | TRA | BBU | AGR | SHP |
| Lelieveld et al. (2015) | Population-weighted | - | **32** | 8 | 18 | 3 | 1 | 29 | - |
| Silva et al. (2016)[a] | Population-weighted | 34.2 | **32** | 26 | 17 | 6 | - | - | - |
| Archer-Nicholls et al. (2016) | Not specified (assume population-weighted) | - | 37 | - | - | - | - | - | - |
| GBD-MAPS (2016)[b] | Population-weighted | 54.3 | 19.2 | **27.3** | 9.4 | 15.0 | 7.6 | - | - |
| Karagulian et al. (2017)[c] | Not specified (assume population-weighted) | 55 | 26.7 | **38.2** | 14.5 | 6.4 | 3.0 | 11.3 | - |
| Hu et al. (2017) | Population-weighted | 62.6 | 21.7 | **30.5** | 10.3 | 5.7 | 4.9 | 12.2 | |
| Aunan et al. (2018)[d] | Population-weighted | 58 | 19.0 | - | - | - | - | - | - |
| Butt et al., *in prep.* (2019) | Population-weighted | - | 34 | - | - | - | - | - | - |
| This study | Population-weighted | 72.3 | 38.1 | **43.1** | 5.3 | 3.8 | 1.0 | 0.1 | 0.1 |
| Butt et al. (2016) | Area-weighted | - | 13 | **-** | - | - | - | - | - |
| Shi et al. (2017)[e] | Area-weighted | - | 18.5 | **26.6** | 9.6 | 4.7 | 6.4 | 10.8 | - |
| Gao et al. (2018)[f] | Area-weighted | - | 24.2 | **35.7** | 33.2 | 6.9 | - | - | - |
| Gu et al. (2018)[g] | Area-weighted | - | 24.9 | **32.0** | 12.8 | 7.3 | - | 15.6 | - |
| This study | Area-weighted | 32.2 | **39.1** | 37.1 | 5.3 | 3.1 | 2.9 | 0.1 | 0.1 |
| Multi-model mean | Population-weighted | 56 | **29** | **29** | 12 | 7 | 4 | 13 | - |
| Multi-model mean | Area-weighted | - | 24 | **33** | 15 | 6 | 5 | 9 | - |
| Multi-model mean | All values | 52 | 26 | **30** | 14 | 7 | 4 | 13 | - |

[a] Relative contributions are for all of East Asia (including China).

[b] Relative contributions calculated using mean values from Table 6 of GBD-MAPS (2016). ENE = Power plant coal; IND = Industrial coal + Non-coal industrial; RES = Domestic coal + Domestic biomass burning.

[c] Relative contributions calculated from national annual mean PM$_{2.5}$ concentrations in Sect. 3.1 of Karagulian et al. (2017). Missing sector for China (open biomass burning) was calculated from the remaining fraction of PM$_{2.5}$ (Table T5 is missing from the report).

[d] Relative contributions calculated from values of "population-weighted exposure to ambient air pollution" in Table 1 of Aunan et al. (2018).

[e] Relative contributions calculated as average fractions across all provinces from Table 3 of Shi et al. (2017).

[f] Relative contributions taken from Fig. S5 of Gao et al. (2018), showing sectoral contributions to area-weighted mean PM$_{2.5}$ concentrations.

[g] Relative contributions for RES, IND and TRA sectors taken from the text (Sect. Impacts on air quality of Gu et al. (2018)) assuming these refer to area-weighted annual mean concentrations. Individual relative contributions for AGR and ENE sectors calculated from combined value in text (28.4%) and relative contributions of population-weighted concentrations in Fig. 2 of Gu et al. (2018).





**Table 5.** Comparison of relative sector-specific contributions to simulated annual mean PM$_{2.5}$ concentrations over India from this study and previous studies. Emission sectors are: agriculture (AGR), biomass burning (BBU), power generation (ENE), industrial non-power (IND), residential energy use (RES), and land transport (TRA). The largest relative contribution for each study is in bold. The average over all studies (multi-model mean) is shown for population-weighted, area-weighted, and all annual mean PM$_{2.5}$ concentrations and relative contributions.

| Reference | Population-weighted or area-weighted annual mean PM$_{2.5}$ | Annual mean PM$_{2.5}$ concentration for India | Relative sector-specific contributions to simulated annual mean PM$_{2.5}$ concentrations (%) | | | | | |
|---|---|---|---|---|---|---|---|---|
| | | | RES | IND | ENE | TRA | BBU | AGR |
| Lelieveld et al. (2015) | Population-weighted | - | **50** | 7 | 14 | 5 | 7 | 6 |
| Silva et al. (2016) | Population-weighted | 28.5 | **43** | 11 | 15 | 7 | - | - |
| Karagulian et al. (2017)[a] | Not specified (assume population-weighted) | 51 | **42** | 18 | 21 | 10 | - | - |
| GBD-MAPS (2018)[b] | Population-weighted | 74.3 | **23.9** | 9.9 | 7.6 | 2.1 | 5.5 | - |
| Guo et al. (2018) | Population-weighted | 32.8 | **55.5** | 19.7 | 6.8 | 1.9 | - | 11.9 |
| Butt et al., *in prep.* (2019) | Population-weighted | - | 28 | - | - | - | - | - |
| This study & Conibear et al. (2018a) | Population-weighted | 57.2 | **51.6** | 16.3 | 21.0 | 10.3 | 2.8 | 0.3 |
| Butt et al. (2016) | Area-weighted | - | 22 | - | **-** | - | - | - |
| Gao et al. (2018)[c] | Area-weighted | - | 23.9 | 16.2 | **40.1** | 19.8 | - | - |
| This study & Conibear et al. (2018a) | Area-weighted | 42.1 | **47.4** | 15.2 | 22.4 | 10.3 | 4.0 | 0.3 |
| Multi-model mean | Population-weighted | 49 | **42** | 14 | 14 | 6 | 5 | 6 |
| Multi-model mean | Area-weighted | - | **31** | 16 | 31 | 15 | - | - |
| Multi-model mean | All values | 49 | **38** | 14 | 18 | 8 | 5 | 6 |

[a] Relative contributions calculated from national annual mean PM$_{2.5}$ concentrations quoted in Sect. 3.1 of Karagulian et al. (2017). Two sectors are missing for India (biomass burning and agriculture) so we were unable to calculate these fractions (Table T5 is missing from the report).

[b] Relative contributions taken from Table 2 of GBD-MAPS (2018).

[c] Relative contributions taken from Fig. S5 of Gao et al. (2018), showing sectoral contributions to national mean PM$_{2.5}$ concentrations. We assume the fraction quoted in the text (32% in India; Gao et al., 2018) is the contribution to the *population-weighted* annual mean PM$_{2.5}$ concentration.





**(a)**

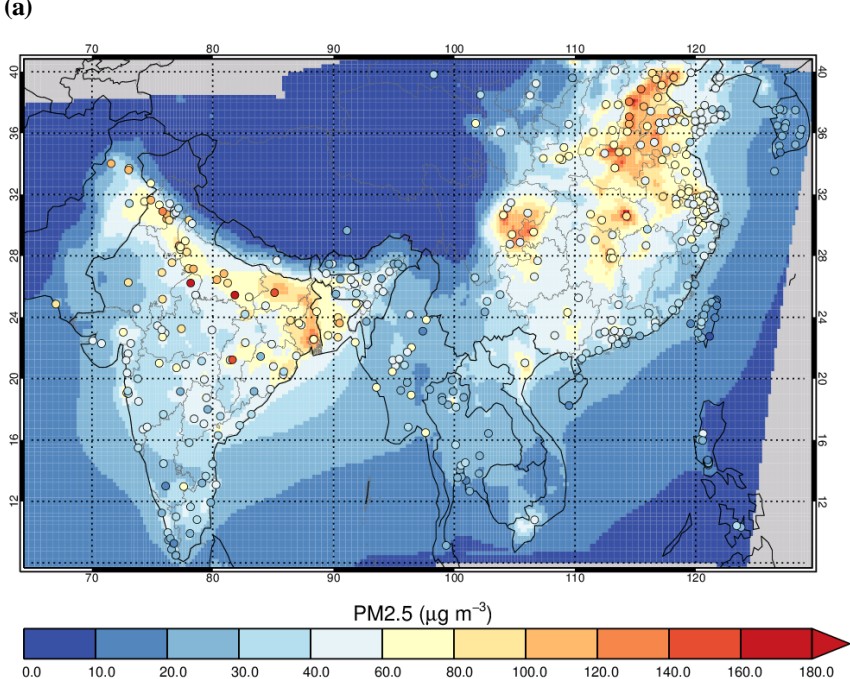

**(b)**

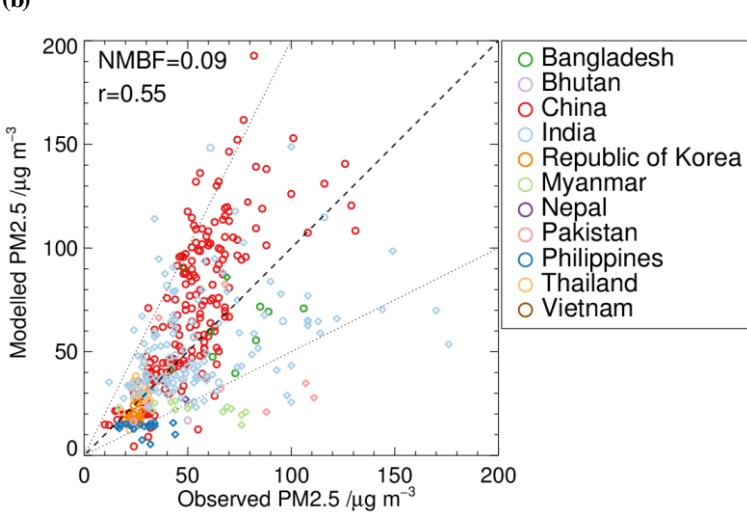

**Figure 1.** Simulated and measured annual mean surface PM$_{2.5}$ concentrations across South and East Asia. Observation data is from the World Health Organization database, 2016 & 2018. **(a)** Map of the simulated surface distribution of annual mean PM$_{2.5}$ for 2014 (underlying colours); overlying circles show measured annual mean PM$_{2.5}$ concentrations for available years (2009-2016). Regions in grey are outside the model domain. **(b)** Simulated versus measured annual mean PM$_{2.5}$ concentrations. Circles show measured annual mean PM$_{25}$ concentrations for the year 2014; diamonds show measured annual mean PM$_{2.5}$ concentrations for years other than 2014. All simulated annual mean PM$_{2.5}$ concentrations are for the year 2014. The normalised mean bias factor (NMBF; Yu et al., 2006) and Pearson's correlation coefficient (r) between simulated and measured values are displayed in the top left corner.





**Figure 2.** Relative contributions of different anthropogenic emission sectors to population-weighted annual mean PM$_{2.5}$ concentration by country in South and East Asia. Emission sectors include: agriculture (AGR), power generation (ENE), industrial non-power (IND), residential energy use (RES), land transport (TRA), open biomass burning (BBU) and shipping (SHP; China and Mainland Southeast Asia only). Where the percentage contributions from each sector do not add up to 100%, the residual fraction is assigned to "Natural and minor sources" (NAT). Relative contribution values of 10% or greater are shown on the quadrants. Results are shown for the region of China contained within the model domain, which accounts for 92% of the Chinese population (Sect 2.2).




**(a)**

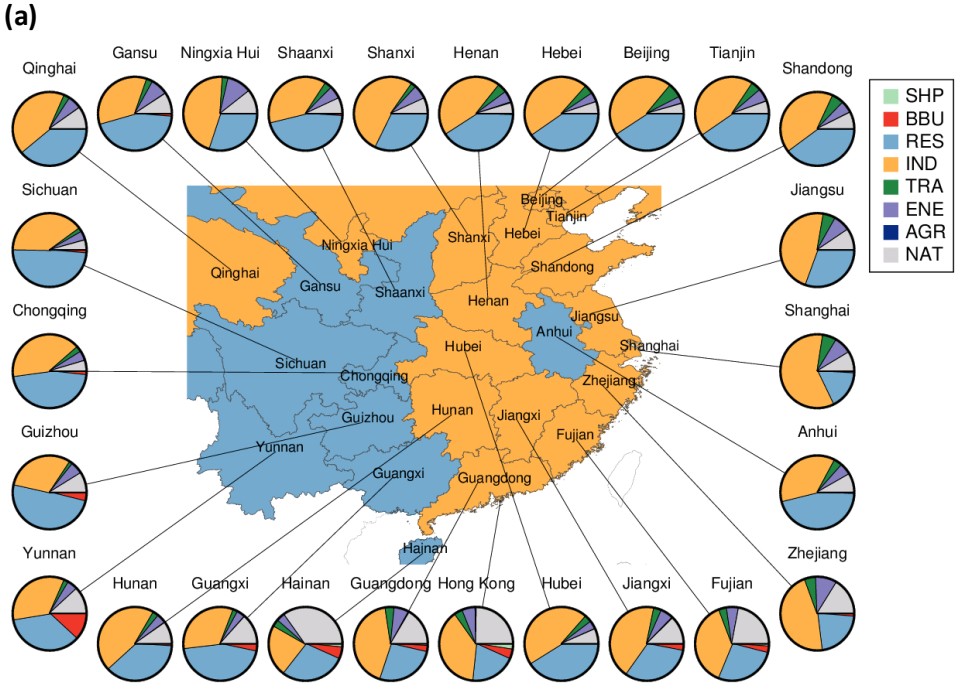

**(b)**

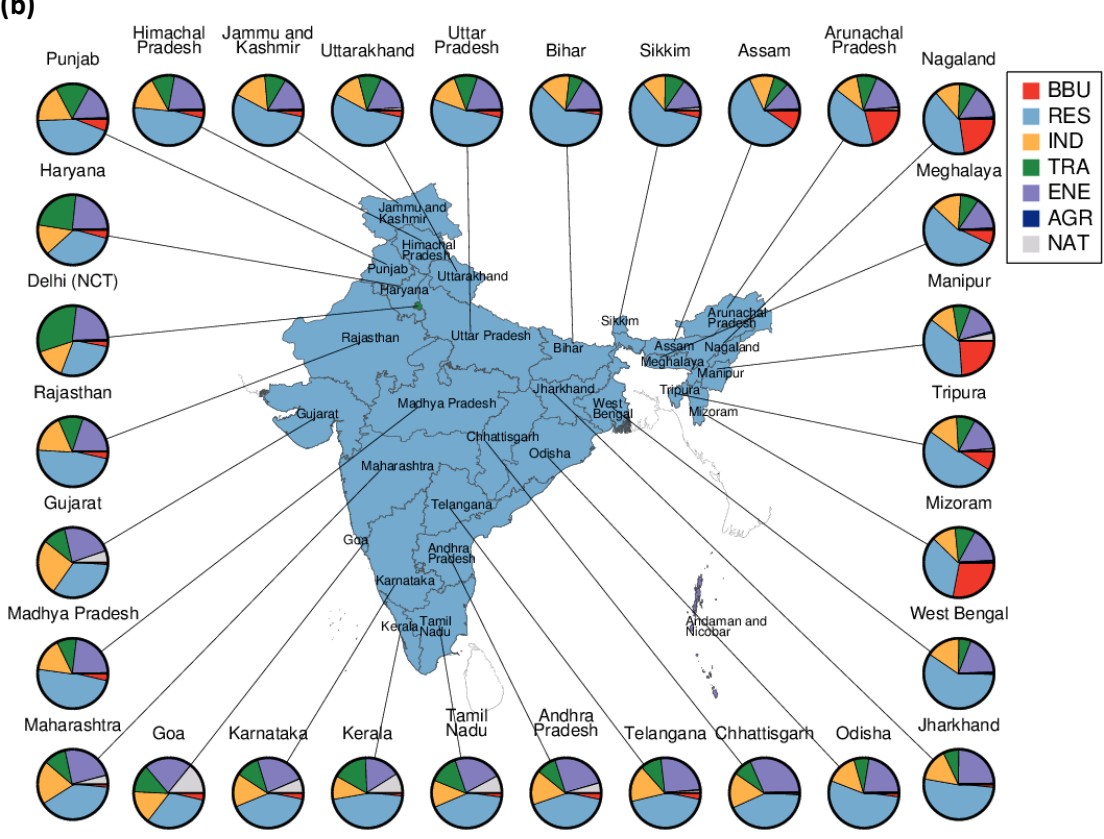



**Figure 3.** Contribution of different emission sectors to population-weighted annual mean PM$_{2.5}$ concentration **(a)** by province/municipality/region in China; and **(a)** by state in India (Union Territories are not shown individually apart from Delhi National Capital Territory (NCT)). The colour of each province in China and each state in India indicates the sector that dominates contributions to population-weighted annual mean PM$_{2.5}$ in that province or state. The emission sectors are: agriculture (AGR), power generation (ENE), industrial non-power (IND), residential energy use (RES), land transport (TRA), open biomass burning (BBU) and shipping (SHP; China only). Where the percentage contributions from each sector do not add up to 100%, the residual fraction is assigned to "Natural and minor sources" (NAT).





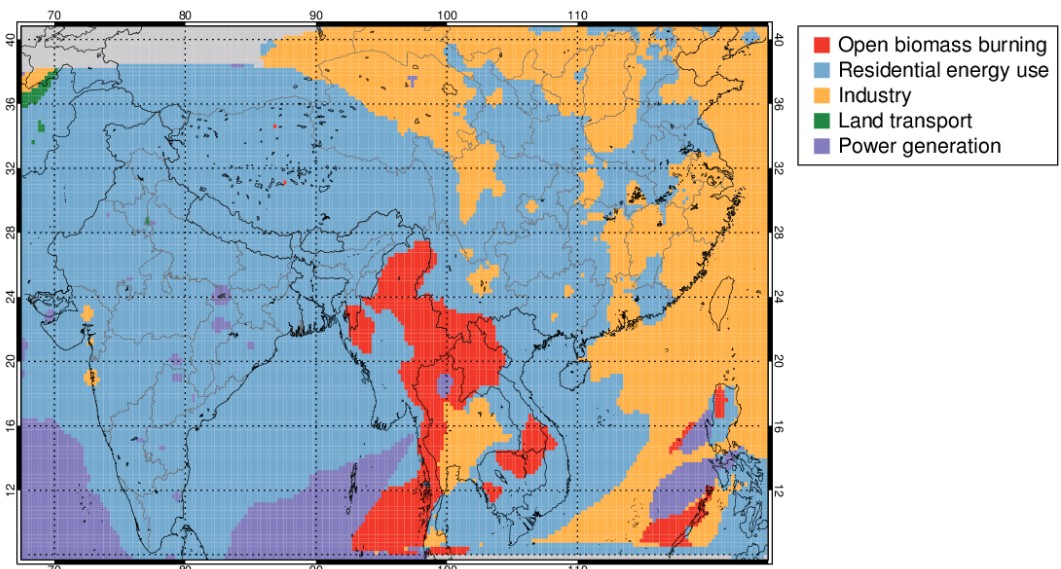

**Figure 4.** Spatial distribution of the dominant anthropogenic emission sectors for annual mean PM$_{2.5}$ in South and East Asia. The dominant emission sector is calculated for each model grid cell as the emission sector that gives the largest reduction in simulated annual mean surface PM$_{2.5}$ concentration i.e. results in the largest absolute difference in µg m$^{-3}$ from the control simulation. Regions in grey are outside the model domain.

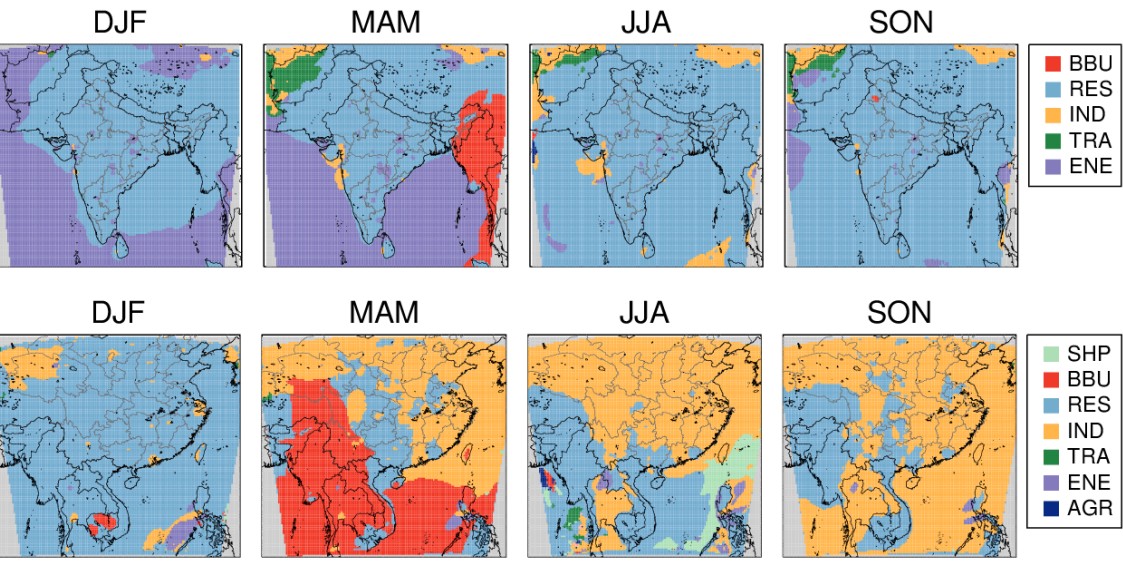

**Figure 5.** Spatial distribution of the dominant anthropogenic emission sectors for seasonal mean PM$_{2.5}$ in South Asia (top panel) and East Asia (bottom panel). DJF = December, January, February mean; MAM = March, April, May mean; JJA = June, July, August mean; SON = September, October, November mean. As for Fig. 4, the dominant emission sector is calculated for each model grid cell as the emission sector that gives the largest reduction in simulated seasonal mean surface PM$_{2.5}$ concentration i.e. results in the largest absolute difference in µg m$^{-3}$ from the control simulation. Regions in grey are outside the model domain. The emission sectors are: agriculture (AGR), power generation (ENE), industrial non-power (IND), residential energy use (RES), land transport (TRA), open biomass burning (BBU) and shipping (SHP; East Asia only).





**Figure 6. (a)** Total annual premature mortality per country due to long-term exposure to ambient PM$_{2.5}$ from all emission sources. The
5   colours show premature mortality by disease (chronic obstructive pulmonary disease (COPD), ischaemic heart disease (IHD), stroke (STR),
lung cancer (LC), and lower respiratory infection (LRI)). **(b)** The number of averted annual premature mortalities due to a reduction in
exposure to ambient PM$_{2.5}$, achieved by eliminating emissions from each sector individually (agriculture (AGR), power generation (ENE),
industrial non-power (IND), residential energy use (RES), land transport (TRA), open biomass burning (BBU) and shipping (SHP; East Asia
only). **(c)** The number of averted annual premature mortalities per 100,000 head of population. Error bars in (a), (b) and (c) represent 95%
10  uncertainty intervals calculated from combining fractional errors in quadrature (see Sect. S1.1 in Supplementary Material). Mortality
estimates for China include Hong Kong SAR, Macau SAR and Taiwan.







**Figure 7.** Comparison of relative sector-specific contributions to annual mean PM$_{2.5}$ concentrations in **(a)** China and **(b)** India from this study and previous studies. Bars show sector contributions to population-weighted annual mean PM$_{2.5}$ concentrations, with the exception of the bars associated with studies shown in the legend with an asterisk, which show estimated sector contributions to surface area-weighted annual mean PM$_{2.5}$ concentrations. In our study, population-weighted and area-weighted values differ by less than six percentage points. The mean relative contribution of each sectors is shown above the bars with the range of values (minimum to maximum) in parenthesis. The values for each study are also shown in Tables 4 and 5. The emission sectors are: agriculture (AGR), power generation (ENE), industrial non-power (IND), residential energy use (RES), land transport (TRA), and open biomass burning (BBU).