# Peer review of "Exploring the impacts of anthropogenic emission sectors on PM2.5 and human health in South and East Asia"

_Atmospheric Chemistry and Physics, 2019_

## Referee Comment (RC1) · Anonymous Referee #2 · 5 Apr 2019

As a reviewer I have the following comments: GENERAL 1. Why does your approach is better than using the measurements from monitor stations? 2. How your results can be used? No just to report the analysis but say in longitudinal studies? 3. You have an implicit assumption that PM2.5s are the same, but their structure/composition, specially urban vs. rural are very different. Probably their health effects also.

SPECIFIC 1. Formula (2). We remove "SECTOR_OFF". It assumed that no pollution, but this sector is replaced by something else, say biomass burning by LPG. 2. You provided interesting example when migration to cities reduces biomass burial – no policy. TECHNICAL 1. Abstract; Line 11. "energy generation" – humans don't generate

energy. We only change. 2. Abstract; Line 16. After "467, 000" you missed "UI", but please full spell in the first time of use. 3. Abstract; Line 13. PM2.5 it includes also =, I think you may say "no greater than 2.5...". Also in the text, definition of PM2.5. 4. Page 3; Line 8: you are using the term "exposure-response" – I think more adequately is to use "concentration-response" (but keep IER as is). We know the ambient concentration levels. 5. Page 3; Line 24: We don't know what "UI" is. 6. Page 3; Line 29: "high resolution" –does it mean very detailed, or "high level"? 7. Page 4; Line 15: The intervals, suggestion, show as the intervals [0.039, 0.156] or ) included vs. non-included. 8. Page 5; Line 31. Please change "Savannah". 9. Page 7; Lines 8-9. "billion" –I know it's milliard. I am guessing that you use the journal locality? 10. Page 9; Line 13. Please define UI in its first occurrence. Thank you

---

## Referee Comment (RC2) · Anonymous Referee #1 · 7 Apr 2019

**Review of "Exploring the impacts of anthropogenic emission sectors on PM$_{2.5}$ and human health in South and East Asia" by Reddington et al.**

The authors estimated the sectoral contributions to PM$_{2.5}$ concentration and its health impacts in Asia, and they also have made a comprehensive comparison of earlies sector-specific emission studies. Air quality issue in Asian countries is of great concern among scientific communities. The topic is well within the scope of this journal. The methodology in this study is solid, and the conclusions are well defended and discussed. I think the paper is publishable after the following comments/suggestions are addressed.

General comments:
-Agricultural contribution. The authors show a very small contribution (~0.1%) from agriculture sector, which is too low compared with previous studies (Fig.7). In addition, earlies studies (e.g., Zhang et al., 2015; Liu et al., 2019) reported the importance of controlling NH$_3$ emissions, about 90% of which source from agricultural activities, to decrease PM$_{2.5}$ concentration in China. I suggest the authors to re-examine this or at least more discussion on such discrepancy.

-Biomass burning contribution. (1) It's reasonable to have a low estimated contribution from biomass burning emissions, because FINN satellite products tend to underestimate agricultural fire emissions in China (Shen et al., 2019). However, the simulated spatial contribution from biomass burning (Fig. S2c) is not very consistent with other studies (e.g., Li et al., 2016). It will be better for readers to understand the numbers in this work if there is more discussion on this. (2) The authors also should note there is strong interannual variations for biomass burning emissions. (3) when saying the contribution from biomass burning, the authors used words like "excluding fire emissions" (P12L5, P12L20). In fact, wildfire emissions are not so controllable as those from anthropogenic fires. Should make it more clear in the text.

Specific comments:
-P5L15. Anthropogenic emissions from HATP are for year 2010. Are all the species compiled from regional emission inventories for year 2010. Please make it clear in this section.
-Correlation between simulated and observed PM$_{2.5}$ in India is only 0.37, though regional mean values are not biased. More discussion on the model capability to simulate PM$_{2.5}$ in India is needed.
-In Fig.4 and Fig.5, I suggest to mask the sectoral contribution over oceans where the value reads kind of weird.
-In Fig.6, Fig.6b and 6c are like almost the same. It is reasonable to show only one (e.g., Fig.6b).
-P38L5: in Fig.7 caption, please add a "(*)" after the text "in the legend with an asterisk". A symbol is more catching than a word in text.

Reference:

1. Li et al. (2016). Source sector and region contributions to concentration and direct radiative forcing of black carbon in China. *Atmos. Environ.* 124, 351-366.  2. Liu et al. (2019). Ammonia emission control in China would mitigate haze pollution and nitrogen deposition, but worsen acid rain. *PNAS*, 201814880.  3.Shen et al. (2019). 2005-2016 trends of formaldehyde columns over China observed by satellites: increasing anthropogenic emissions of volatile organic compounds and decreasing agricultural fire emissions. Geophys. Res. Lett.  4.Zhang et al. (2015). Source attribution of particulate matter pollution over North China with the adjoint method. *Environ. Res. Lett.* 10(8):084011.

---

## Author Comment (AC1) · 30 Jul 2019

**Author response to referees**

We would like to thank the referees for taking time to review our manuscript and for all the insightful comments they have provided. We have responded to all the referee comments below and have modified our manuscript accordingly. We have included additional co-authors (Chak K. Chan and Yong Jie Li) to the revised paper for providing aerosol composition measurement data with which to evaluate our model simulations with. Our manuscript has been strongly improved through the review process and we hope it is now suitable for publication.

To guide the review process, referee comments below are in plain text and our responses are in italics, additions to our manuscript are shown below in red and as yellow highlighted sections in the revised manuscript.

**Anonymous Referee #1**

The authors estimated the sectoral contributions to PM2.5 concentration and its health impacts in Asia, and they also have made a comprehensive comparison of earlies sector-specific emission studies. Air quality issue in Asian countries is of great concern among scientific communities. The topic is well within the scope of this journal. The methodology in this study is solid, and the conclusions are well defended and discussed. I think the paper is publishable after the following comments/suggestions are addressed.

*Thank you for the positive comments on our manuscript.*

**General comments:**
  1. Agricultural contribution. The authors show a very small contribution (~0.1%) from agriculture sector, which is too low compared with previous studies (Fig.7). In addition, earlies studies (e.g., Zhang et al., 2015; Liu et al., 2019) reported the importance of controlling NH3 emissions, about 90% of which source from agricultural activities, to decrease PM2.5 concentration in China. I suggest the authors to re-examine this or at least more discussion on such discrepancy.

     *We agree that our calculated contribution of the agricultural sector to PM2.5 is very small relative to other studies (e.g. Karagulian et al., 2017; Hu et al., 2017; Shi et al.; 2017). As noted in the paper, there have been fewer studies quantifying the contribution of agriculture to PM2.5 concentrations in China and India, and the contribution of this sector has the largest uncertainty. Therefore we also agree that a more detailed evaluation of our model results was necessary.*

     *We used a collection of aerosol mass spectrometer measurements (from Li et al. 2017; described in new Table S1 in the supplementary material) to evaluate speciated aerosol mass concentrations in our model. This evaluation showed that the model simulates organic and sulphate aerosol concentrations reasonably well (within a factor 2) but underestimates nitrate and ammonium concentrations (see new Section 3.1.2 and new Fig. S2). An underestimation of measured ammonium concentrations suggests that our estimate of the contribution of the agricultural sector (NH₃ emissions) to PM2.5 is underestimated. Therefore, we have decided to remove the analysis and contribution of this sector from our paper, instead focussing on six emission sectors (power generation, industrial non-power, residential energy use, land transport, open biomass burning, and shipping).*

     *We have also added the following text to Sect. 4 discussing the contribution of agriculture to PM2.5 found in other studies:*

     "We have not quantified the contribution of the agricultural sector to PM2.5 in China. Our model simulations underestimate ammonium concentrations over China (Sect. 3.1.2)

and are therefore it is likely that we would underestimate the contribution of the agriculture sector to PM2.5 concentrations. Previous studies have found this sector contributes as much as 11-29% (mean 16%; Table 4) in China and 0.3-12% (mean 6%; Table 5) in India to annual mean PM2.5 concentrations. There have been fewer studies quantifying the contribution of agriculture to PM2.5 concentrations in China and India relative to the other emission sectors, and the contribution of this sector has large uncertainty. Future work requires a detailed comparison of simulated and observed composition resolved aerosol mass to help inform these sector-based emission studies."

2. Biomass burning contribution.

(1) It's reasonable to have a low estimated contribution from biomass burning emissions, because FINN satellite products tend to underestimate agricultural fire emissions in China (Shen et al., 2019). However, the simulated spatial contribution from biomass burning (Fig. S2c) is not very consistent with other studies (e.g., Li et al., 2016). It will be better for readers to understand the numbers in this work if there is more discussion on this.

*We agree that the FINN fire emissions are likely to underestimate agricultural fire emissions in China and we have acknowledged this in the main text (Sect 3.2.1):*

*"…it is likely that fire emission datasets underestimate the emissions from agricultural fires in China (e.g. Zhang et al., 2016)…"*

*We have now included an additional reference in the above sentence; citing the Shen et al. (2019) study. We have also included text to the conclusions further acknowledging the likely underestimation of agriculture fire emissions by the FINN dataset (please see response to referee comment (3) below).*

*With regard to the differences between Fig. S2c and Figure 7 in Li et al. (2016): we would expect differences between the spatial contribution from biomass burning to annual mean PM2.5 in 2014 (our study) and the spatial contribution from biomass burning to seasonal mean black carbon (BC) concentrations in 2010 (Li et al., 2016), due to the differences in the: fire emissions (2014 versus 2010), aerosol components (PM2.5 versus BC), and averaging periods (annual versus seasonal), with further differences due to using different models (Geos-Chem versus WRF-Chem), spatial resolutions and fire emissions inventories. We note that the change in annual mean PM2.5 concentrations due to switching off fire emissions in our study (and thus the change in premature mortality shown in Figure S2) is dominated by the reduction in organic aerosol (rather than BC).*

(2) The authors also should note there is strong interannual variations for biomass burning emissions.

*This is a good point. We have now added a sentence to the conclusions of the revised manuscript to acknowledge this (please see response to referee comment (3) below).*

(3) When saying the contribution from biomass burning, the authors used words like "excluding fire emissions" (P12L5, P12L20). In fact, wildfire emissions are not so controllable as those from anthropogenic fires. Should make it more clear in the text.

*This is a good point. We note that the highest particulate fire emissions over the region of interest are likely dominated by manmade agricultural or deforestation fires (see inserted figures below). However, we acknowledge that wildfires will also contribute to particulate fire emissions over the region and are more difficult to control than agricultural or deforestation fires.*

[Figure]

The figures above show dominant fire type (DEFO = deforestation, AGRI = agriculture, SAVA = savannah, TEMF = temperate forest) (left) and mean biomass burning emissions of black carbon (BC) aerosol averaged over 2002-2015 over East Asia (right). Data is from GFED4 (van der Werf, 2017).

*In response to referee comments (1)-(3) above, we have added the following paragraph to the Conclusions (Sect. 5) in the revised manuscript:*

"Effective options also exist within the agricultural sector to reduce emissions from open biomass burning and improve air quality, including "no burn" alternatives to clearing agricultural residues and/or stricter enforcement of bans on open burning. The occurrence of wildfires is more difficult to control but may be reduced by improving forest and land management and by employing fire prevention strategies. Emissions from agricultural fires are likely underestimated in China, India and Southeast Asia by the fire emissions dataset used in this study and so open biomass burning may make a larger contribution to PM2.5 concentrations than reported here. Open biomass burning emissions in some regions in Asia show strong inter-annual variation and so contributions to PM2.5 concentrations may vary from year to year. The contribution of open biomass burning to air pollutant concentrations in Asia should be analysed in detail in future work; using additional observations for model constraint."

**Specific comments:**

1. P5L15. Anthropogenic emissions from HATP are for year 2010. Are all the species compiled from regional emission inventories for year 2010. Please make it clear in this section.

   *The regional emissions inventories included in the MIX mosaic inventory (which is used for emissions over Asia in the EDGAR-HTAP2 emission inventory) are mostly developed for the year 2010, including the Multi-resolution Emission Inventory for China (MEIC) for 2010, Indian emissions from the Argonne National Laboratory for 2010, and gap-filling from REAS2.1 for 2010. The NH3 emission inventory for China from Peking University was developed for the year 2006. We have now inserted these years into the text of Sect. 2.1.1.*

2. Correlation between simulated and observed PM2.5 in India is only 0.37, though regional mean values are not biased. More discussion on the model capability to simulate PM2.5 in India is needed.

   *In comparison to WHO measurements, we agree that the spatial correlation between simulated and measured PM2.5 in India is low relative to the model-measurement comparison in China. This is likely to be mainly due to the large range in measurement years for WHO urban PM2.5 measurements in India (2012-2016), with only 11 stations with measurements available for 2014 (the simulation year) and no available measurements for 2010 (the year of the emissions inventory used). Comparing*

*simulated to PM2.5 against measurements from 2014 only (11 stations), we obtain improved spatial correlation and bias between model and measurements (r=0.67, NMBF= -0.01).*

*We have now added the following text to Sect. 3.1.1:*

"In India, the model is generally unbiased against the measurements (NMBF=-0.05), as reported by Conibear et al. (2018a) who used Central Pollution Control Board (CPCB) measurement data for 2016 to evaluate simulated $PM_{2.5}$ concentrations. The spatial correlation between simulated and measured annual mean $PM_{2.5}$ in India (r=0.37; Table 1) is low relative to the model-measurement comparison in China (r=0.76). We suggest this is mainly due to the large range in measurement years for the WHO $PM_{2.5}$ measurements in India (2012-2016; Table 1), with only 11 stations with measurements available for 2014 (the simulation year) and no available measurements for 2010 (the year of the emissions inventory used). Comparing simulated annual mean $PM_{2.5}$ against measurements from 2014 only (11 stations), we obtain improved spatial correlation and bias between model and measurements (r=0.67, NMBF= -0.01)."

*We note that the published paper Conibear et al. (2018) includes a more detailed evaluation of simulated PM2.5 and aerosol optical depth over India (using an identical model set-up).*

3. In Fig.4 and Fig.5, I suggest to mask the sectoral contribution over oceans where the value reads kind of weird.

   *These figures show the sectoral contributions to surface PM2.5 concentrations. The contributions over the oceans are valid. We acknowledge that there may be less interest in source contributions over the ocean. However, the plots do indicate outflow of pollution and so we prefer to maintain the plots as they are.*

4. In Fig.6, Fig.6b and 6c are like almost the same. It is reasonable to show only one (e.g., Fig.6b).

   *We agree that there are some similarities between Fig. 6b (number of averted mortalities) and Fig. 6c (mortality rate). However, there are also important differences. For example, whilst the number of avoided mortalities is much higher in India and China, the averted mortality rate in India and China is similar to the other countries studied. For this reason we prefer to retain both figures in the main paper.*

5. P38L5: in Fig.7 caption, please add a "(*)" after the text "in the legend with an asterisk". A symbol is more catching than a word in text.

   *We agree. Now added.*

Reference:

1. Li et al. (2016). Source sector and region contributions to concentration and direct radiative forcing of black carbon in China. Atmos. Environ. 124, 351-366.

2. Liu et al. (2019). Ammonia emission control in China would mitigate haze pollution and nitrogen deposition, but worsen acid rain. PNAS, 201814880.

3. Shen et al. (2019). 2005-2016 trends of formaldehyde columns over China observed by satellites: increasing anthropogenic emissions of volatile organic compounds and decreasing agricultural fire emissions. Geophys. Res. Lett.

4. Zhang et al. (2015). Source attribution of particulate matter pollution over North China with the adjoint method. Environ. Res. Lett. 10(8):084011.

**Anonymous Referee #2**

As a reviewer I have the following comments:

**GENERAL**

1. Why does your approach is better than using the measurements from monitor stations?

   *The main aim of our study is to provide information on the contribution of different emission sectors to particulate matter concentrations. The monitoring stations provide information on the concentrations of different pollutants, but can't provide information on the contribution of different emission sectors to observed concentrations. Our modelling approach has several additional advantages over using only measurements from monitoring stations, such as enabling us to:*
   - *Examine PM2.5 concentrations at locations where no observed data is available, providing PM2.5 concentrations over regions where no monitoring stations exist.*
   - *Test the impacts and results of planned or hypothetic emissions control scenarios before they have been implemented in the real world.*
   - *Quantify pollution-source contributions in detail at different times of the year and over large regions.*

   *We note that it is important to combine the modelling approach with validation of the simulated pollutant concentrations against measurements from monitoring stations as we have done in our study.*

2. How your results can be used? No just to report the analysis but say in longitudinal studies?

   *The results from this study can be used to inform effective emission-reduction strategies at the local level across South and East Asia to improve air quality and reduce the substantial disease burden from air pollution exposure. In the conclusions we use our results to recommend that emission-reduction strategies in India, China and Mainland Southeast Asia should focus on reducing the combustion of solid fuels in homes, industry, and through open burning.*

3. You have an implicit assumption that PM2.5s are the same, but their structure/composition, specially urban vs. rural are very different. Probably their health effects also.

   *We agree that the structure and composition (and contributing sources) of PM2.5 varies over the region of interest in our study. The WRF-Chem model used in our study simulates the composition and physical properties of aerosol particles at 30 km horizontal resolution. Thus the model resolves broad changes in PM2.5 composition between rural/background regions and urban-dominated regions within the model domain, which is taken into account when quantifying the major contributing sources to PM2.5.*

   *We have now added an evaluation of simulated aerosol chemical composition (see new Sect. 3.1.2 and Fig. S2) to the revised manuscript and we have added the following text to the model description (Sect. 2.1):*

   "The MOSAIC scheme treats major aerosol species including sulphate, nitrate, chloride, ammonium, sodium, black carbon, primary and secondary organic aerosol and other inorganics (including crustal and dust particles and residual primary PM$_{2.5}$)."

   *The reviewer raises a good point regarding the role of chemical composition on human health. Treating the toxicity of PM2.5 as homogenous in terms of structure and chemical composition is consistent with the Global Burden of Disease (GBD) Project and numerous other recent health impact studies cited in our manuscript (e.g. Archer-Nicholls et al., 2016; Hu et al., 2017; Conibear et al., 2018; Upadhyay et al., 2018).*

*Although links have been found between PM2.5 composition and health effects, there is currently insufficient long-term measurements of aerosol composition, particularly in Asia, to carry out adequate health impact assessments of specific aerosol species or mixtures. Therefore, there are currently no aerosol composition-dependent exposure-response functions available. We have added the following text to Sect. 2.2 of the revised manuscript:*

"The toxicity of PM2.5 is treated as homogenous toxic regarding source, shape, and chemical composition, consistent with the GBD project, due to lack of composition-dependent exposure-response functions."

**SPECIFIC**

1. Formula (2). We remove "SECTOR_OFF". It assumed that no pollution, but this sector is replaced by something else, say biomass burning by LPG.

   *Yes, when we remove each individual emission sector in the model, we assume that pollution is no longer emitted from that specific source. We agree that in reality this source may be replaced by another pollution source, but we have not tested this scenario in our study. We have added the following text to Sect. 2.1 to clarify this:*

   "When the emission sector is switched off in the model, pollution is no longer emitted from that specific source. In reality, the removed emission sector may be replaced by another pollution source but this scenario is not tested in this study."

2. You provided interesting example when migration to cities reduces biomass burial – no policy.

   *We thank the reviewer for this comment.*

**TECHNICAL**

1. Abstract; Line 11. "energy generation" – humans don't generate energy. We only change.

   *We have now replaced this phrase with* *"electricity generation"*.

2. Abstract; Line 16. After "467, 000" you missed "UI", but please full spell in the first time of use.

   *Thank you. We have now corrected this mistake and added the definition for "95UI":* "95% uncertainty interval (95UI)".

3. Abstract; Line 13. PM2.5 it includes also =, I think you may say "no greater than 2.5: : :". Also in the text, definition of PM2.5.

   *As suggested, we have replaced this text in the abstract and introduction section with the following definition:*
   "particulate matter with aerodynamic diameter no greater than 2.5 µm"

4. Page 3; Line 8: you are using the term "exposure-response" – I think more adequately is to use "concentration-response" (but keep IER as is). We know the ambient concentration levels.

   *We have now changed "exposure-response" to* *"concentration-response"* *throughout the revised manuscript.*

5. Page 3; Line 24: We don't know what "UI" is.

   *We have now inserted the following definition here:* "95% uncertainty interval (95UI)".

6. Page 3; Line 29: "high resolution" –does it mean very detailed, or "high level"?

   *Here we refer to the detailed spatial resolution of the regional WRF-Chem model used in our study compared to the relatively coarse resolution global and regional models employed by many previous studies. We agree this could be made clearer and have changed the phrase to the following:* "high spatial resolution".

7. Page 4; Line 15: The intervals, suggestion, show as the intervals [0.039, 0.156] or ) included vs. non-included.

   *All sectional bins are included in our model set-up. We have modified the sentence to make it clearer:*

   "Four discrete size bins are used within MOSAIC to represent the aerosol size distribution (with the following dry particle diameter ranges: 0.039–0.156 µm, 0.156–0.625 µm, 0.625–2.5 µm, and 2.5–10 µm)."

8. Page 5; Line 31. Please change "Savannah".

   *We have now changed this to:* "savannah/grassland".

9. Page 7; Lines 8-9. "billion" –I know it's milliard. I am guessing that you use the journal locality?

   *Yes here we use the term "billion" to mean "one thousand million" or "milliard". The usage of the term "milliard" is no longer common in British English and is not used in American English. Therefore, we have chosen to use the widely used and official term of "billion".*

10. Page 9; Line 13. Please define UI in its first occurrence. Thank you.

   *Thank you. We have now defined "UI" in the abstract and introduction.*

**References used in the author response**

Archer-Nicholls, S., Carter, E., Kumar, R., Xiao, Q., Liu, Y., Frostad, J., Forouzanfar, M.H., Cohen, A., Brauer, M., Baumgartner, J., and Wiedinmyer C.: The regional impacts of cooking and heating emissions on ambient air quality and disease burden in China, Environ. Sci. Technol., 50, 9416-9423, 2016.

Conibear, L., Butt, E. W., Knote, C., Arnold, S. R., and Spracklen, D. V.: Residential energy use emissions dominate health impacts from exposure to ambient particulate matter in India, Nat. Commun., 9, 617, https://doi.org/10.1038/s41467-018-02986-7, 2018.

Hu, J., Huang, L., Chen, M., Liao, H., Zhang, H., Wang, S., Zhang, Q., and Ying, Q.: Premature mortality attributable to particulate matter in China: source contributions and responses to reductions, Environ. Sci. Technol., 51, 9950-9959 (2017).

Karagulian, F., Van Dingenen R., Belis C.A., Janssens-Maenhout G., Crippa M., Guizzardi D., and Dentener F.: Attribution of anthropogenic PM2.5 to emission sources, EUR 28510 EN, doi 10.2760/344371, 2017.

Shi, Z., Li, J., Huang, L., Wang, P., Wu, L., Ying, Q., Zhang, H., Lu, L., Liu, X., Liao, H., Hu, J.,: Source apportionment of fine particulate matter in China in 2013 using a source-oriented chemical transport model, Sci. Total Environ., 601, 1476–1487, 2017.

Upadhyay, A., Dey, S., Chowdhury, S. and Goyal, P.: Expected health benefits from mitigation of emissions from major anthropogenic PM2.5 sources in India: Statistics at state level,

Environmental Pollution 242, 1817-1826, https://doi.org/10.1016/j.envpol.2018.07.085 (2018).

van der Werf, G. R., Randerson, J. T., Giglio, L., van Leeuwen, T. T., Chen, Y., Rogers, B. M., Mu, M., van Marle, M. J. E., Morton, D. C., Collatz, G. J., Yokelson, R. J., and Kasibhatla, P. S.: Global fire emissions estimates during 1997–2016, Earth Syst. Sci. Data, 9, 697-720, https://doi.org/10.5194/essd-9-697-2017, 2017.